# Unified Vision–Language Modeling via Concept Space Alignment

**Yifu Qiu**[*]
University of Edinburgh
yifu.qiu@ed.ac.uk

**Paul-Ambroise Duquenne**
FAIR at Meta
padqn@meta.com

**Holger Schwenk**
FAIR at Meta
schwenk@meta.com

## Abstract

We introduce V-SONAR, a vision–language embedding space extended from the text-only embedding space SONAR (Omnilingual Embeddings Team et al., 2026), which supports 1500 text languages and 177 speech languages. To construct V-SONAR, we propose a post-hoc alignment pipeline that maps the representations of an existing vision encoder into the SONAR space. We thoroughly evaluate V-SONAR and show that its embeddings achieve competitive performance on text-to-video retrieval. Equipped with the OmniSONAR text decoder, V-SONAR further surpasses state-of-the-art vision–language models on video captioning tasks, including Dream-1k (Bleu 23.9 vs. 19.6) and PE-Video (Bleu 39.0 vs. 30.0).

Leveraging V-SONAR, we first demonstrate that the Large Concept Model (LCM; LCM team et al. 2024) operating in SONAR and trained with English text only, can perform both single- and multi-visual concept understanding in a zero-shot manner. Finally, we introduce V-LCM, which extends the LCM with vision–language instruction tuning. V-LCM encodes vision and language inputs into an unified sequence of latent embeddings via V-SONAR and SONAR, and it is trained with the same latent diffusion objective for next-embedding prediction as in LCM's text-only pre-training. Experiments on a large-scale multilingual and -modal instruction–tuning data mixture highlight the potential of V-LCM: V-LCM matches state-of-the-art vision-language models on tasks covering image/video captioning and question answering, while significantly outperforming them across 61 rich- to low-resource languages out of all 62 tested languages.

## 1 Introduction

Language- and modality-agnostic embedding spaces have emerged as a powerful paradigm for multilingual and –modal representation learning (Artetxe & Schwenk, 2019; Feng et al., 2020; Ni et al., 2021; Duquenne et al., 2023; Wang et al., 2024b; Chen et al., 2024a). Such spaces have achieved state-of-the-art performance across a wide range of applications, e.g., bitext mining (Schwenk et al., 2019; NLLB Team et al., 2022; Ramesh et al., 2022), and speech–text mining (Duquenne et al., 2021b). Beyond these, embedding spaces with the encoder–decoder architecture such as SONAR (Duquenne et al., 2023) have further enabled generative modeling directly in the latent embedding space. The Large Concept Model (LCM; LCM team et al. 2024) extends this direction by showing that diffusion-based language modeling can operate directly in the language-agnostic latent space, i.e., over continuous embeddings rather than discrete tokens. Despite these advances, existing embedding spaces remain restricted to text and speech, limiting their potential for vision–language tasks.

In this work, we introduce V-SONAR, which extends OmniSONAR (Omnilingual Embeddings Team et al., 2026) to the image and video modality. To the best of our knowledge, this makes OmniSONAR the most universal embedding space covering four modalities[1] and up to 1500 languages. We use teacher-student training (Reimers & Gurevych, 2020; Duquenne et al., 2021a; Heffernan et al., 2022) to align the representations of a state-of-the-art vision encoder, Perception Encoder (Bolya et al., 2025), with SONAR's semantic space in a post-hoc manner. The alignment follows a coarse-to-fine

---

[*]This work was done during an internship of Yifu Qiu at FAIR at Meta.

[1]OmniSONAR supports text in 1.5k languages, speech in 177 languages and the added image and video modalities.

curriculum, over three stages of vision captioning data: (1) large-scale image–caption pairs (12M) for coarse grounding, (2) synthetic video–caption pairs (2M) for temporal adaptation, and (3) high-quality human-annotated video captions (200K) for fine-grained alignment. We evaluate V-SONAR extensively. On zero-shot video retrieval, it achieves Recall@1 of 73.03 on PE-VIDEO, largely surpassing SigLIP2-g-opt (63.91). On zero-shot video captioning, it outperforms state-of-the-art vision–language models, improving BLEU by +18, +4.3 on PE-VIDEO, DREAM-1K, respectively, over the Perception Language Model (Cho et al., 2025).

By aligning V-SONAR to SONAR, we show that the latent diffusion language model operating in SONAR, LCM (LCM team et al., 2024) trained with English textual corpus, can zero-shot process the visual embeddings encoded by V-SONAR. In the single-concept understanding task, i.e., video captioning, LCM only lags behind the existing VLMs with limited margins across PE-VIDEO, DREAM-1K, and VATEX. Similarly, LCM remains competitive for multi-concept reasoning task, i.e., long video summarization as evaluated on VIDEOXUM.

From the view of vision-language modeling, LCM serves as a new paradigm which unifies vision and language modality to the modality-agnostic latent space shared by SONAR and V-SONAR, and directly predict the next embedding with the latent diffusion objective. Therefore, we further introduce a vision-language instruction fine-tuned LCM as an exploration to maximize its utility in various downstream vision-language tasks, named V-LCM. V-LCM encodes multimodal data (images, videos, and text) with V-SONAR and SONAR, and it is trained with the same latent diffusion strategy, following the original two-tower LCM framework (LCM team et al., 2024) in its textual pre-training.

We evaluate V-LCM on the multilingual and -modal instruction-tuning dataset, M3IT (Li et al., 2023), which spans 8 task categories, supports both image and video modalities, and covers 80 languages. V-LCM achieves competitive performance with other vision-language models such as InternVL (Chen et al., 2024b), Qwen-VL (Wang et al., 2024c; Bai et al., 2025) and Perception LM on image/video captioning, visual question answering, and other generation tasks. Notably, in M3IT's multilingual evaluation, V-LCM outperforms other VLMs in 61 languages out of 62 tested languages, ranging from high-resource to low-resource setting. The contributions of this work are:

- We introduce V-SONAR, the first extension of a language- and modality-agnostic embedding space (SONAR) to image and video, via a post-hoc coarse-to-fine alignment strategy.
- We demonstrate that V-SONAR achieves state-of-the-art zero-shot performance on video retrieval and captioning, and generalizes robustly to multilingual settings.
- We show that the LCM, originally trained on text-only data, can effectively operate on V-SONAR embeddings for zero-shot single- and multi-concept vision understanding tasks.
- We extend LCM into a latent diffusion vision–language model (V-LCM) by unifying vision and language in the shared latent space of V-SONAR and SONAR. On M3IT, V-LCM matches state-of-the-art VLMs in captioning and question answering while outperforming them in 61 non-English languages.

## 2 V-SONAR

We begin by introducing V-SONAR, a vision–language embedding space constructed by post-hoc aligning a state-of-the-art vision encoder, PERCEPTION ENCODER, with the multilingual textual embedding space SONAR. We select the PERCEPTION ENCODER as the base encoder for two key reasons: (1) it achieves state-of-the-art performance across both image and video modalities (Bolya et al., 2025), and (2) it has been pre-trained in conjunction with a lightweight text encoder, which facilitates much easier post-hoc alignment with SONAR. This design choice distinguishes PERCEPTION ENCODER from alternative vision encoders such as v-JEPA (Bardes et al., 2023; Assran et al., 2025) and DINO (Oquab et al., 2023; Siméoni et al., 2025), which primarily prioritize visual feature learning without explicit consideration of textual alignment.

**Architecture**   The architecture of V-SONAR is illustrated in the left panel of Figure 1. Given the input image or video, PERCEPTION ENCODER (PE) will first encode each frame separately. Then, we stack a lightweight projector on top of PE to adapt the encoder's representations into the SONAR space. The projector first injects positional embeddings to the embeddings of all frames, thus encoding temporal order information, followed by a single temporal attention layer that enables frame-level

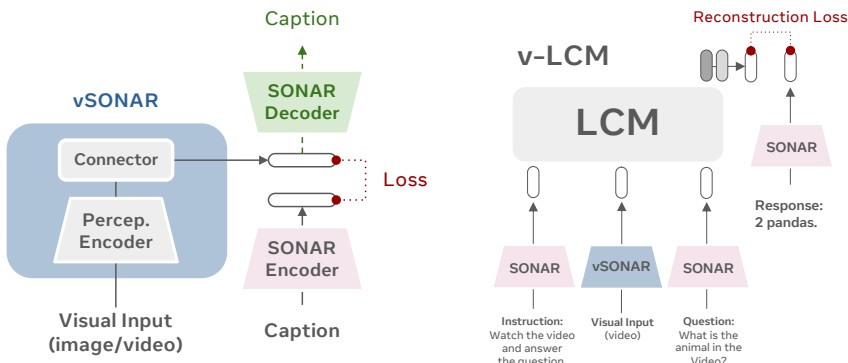

Figure 1: Left: Illustration of V-SONAR. Right: training V-LCM with vision-language instruction tuning.

interactions. Finally, an attention layer then aggregates the frame embeddings into a single video-level representation, which serves as the final embedding for downstream tasks. See Appendix D for implementation details.

**Alignment from Vision to Language.** We use the captioning data for the PERCEPTION ENCODER to align with SONAR, with assumption that the visual inputs and caption should have the same semantic meaning, thus the high-level representations should be as close as possible in the latent modality-agnostic space.

Therefore, given a set of $N$ paired visual inputs and captions $\mathcal{D} = \{(V_i, T_i)\}_{i=1}^{N}$, where $V_i$ is an image or video and $T_i$ is its corresponding caption, we seek to learn a mapping such that the visual embedding $\mathbf{z}_v = f_\theta(V_i)$ and the textual embedding $\mathbf{z}_t = g(T_i)$ share the same semantic space, where $f_\theta$ denotes the trainable vision encoder and $g$ is the frozen SONAR text encoder. To enforce semantic alignment, we minimize the discrepancy between visual and textual embeddings in the SONAR space using Mean Squared Error (MSE) loss:

$$\mathcal{L}_{\text{align}} = \frac{1}{N} \sum_{i=1}^{N} \|f_\theta(V_i) - g(T_i)\|_2^2. \tag{1}$$

Following the teacher-student training (Reimers & Gurevych, 2020; Duquenne et al., 2023), SONAR is frozen and we only update the parameters in the lightweight projector, and the vision encoder. We also experimented with an additional contrastive loss (Oord et al., 2018; Radford et al., 2021) but found no significant gains; details and results are in Appendix B.

We design a coarse-to-fine curriculum to progressively adapt the vision encoder to more complex semantics. The alignment proceeds through three stages. In Stage 1, we initialize alignment using 12M large-scale image–caption pairs from the PLM data pipeline (Cho et al., 2025) which consists of Segment-Anything (Kirillov et al., 2023) and OpenImages (Kuznetsova et al., 2020). This stage establishes a basic mapping between visual and textual embeddings. Then, we introduce 2M pairs from PLM's synthetic

| model | XSIM ↓ | XSIM++ ↓ |
|---|---|---|
| SONAR1 | 1.37 | 15.27 |
| OMNISONAR | 0.65 | 6.14 |

Table 1: Similarity search over 200 languages in FLORES.

video captioning data from YouTube1B corpus (Cho et al., 2025). This step adapts the vision encoder to temporal dynamics while maintaining semantic consistency with SONAR. Finally, we refine the alignment using 200K high-quality human-checked video–caption pairs sourced from PE-VIDEO (Bolya et al., 2025).

We use two versions of the SONAR encoder: SONAR1 is the published and open-sourced version (Duquenne et al., 2023). This is the version supported by the LCM. We had early access to an improved version, named OMNISONAR, which was trained on more data and adds three stages of contrastive training and self distillation (Omnilingual Embeddings Team et al., 2026). As summarized in Table 1, OMNISONAR substantially outperforms SONAR1 on the proxy metric of multilingual

| | Method | R@1$^\uparrow$ | R@5$^\uparrow$ | R@10$^\uparrow$ | MRR$^\uparrow$ | AC$^\uparrow$ | V. Trace$^\uparrow$ | V. logdet$^\uparrow$ | T. Trace$^\uparrow$ | T. logdet$^\uparrow$ |
|---|---|---|---|---|---|---|---|---|---|---|
| PE-Vid | SigLIP2-G-OPT | 47.55 | 71.47 | 79.41 | 58.47 | 0.396 | 0.393 | $-1.7 \times 10^4$ | 0.582 | $-1.7 \times 10^4$ |
| | PECoreG | 63.91 | 85.98 | 91.61 | 73.77 | 0.476 | 0.479 | $-1.4 \times 10^4$ | 0.686 | $-1.4 \times 10^4$ |
| | V-SONAR | **73.03** | **89.75** | **93.81** | **80.50** | **0.519** | **0.700** | $-1.2 \times 10^4$ | **2.216** | $-8.0 \times 10^3$ |
| DREAM | SigLIP2-G-OPT | 61.50 | 83.50 | 89.10 | 71.50 | 0.263 | 0.401 | $-1.8 \times 10^4$ | 0.662 | $-1.8 \times 10^4$ |
| | PECoreG | **72.10** | **89.80** | **93.60** | **79.90** | 0.307 | 0.495 | $-1.4 \times 10^4$ | 0.639 | $-1.4 \times 10^4$ |
| | V-SONAR | 63.30 | 84.10 | 89.00 | 72.46 | **0.410** | **0.559** | $-1.2 \times 10^4$ | **2.523** | $-8.5 \times 10^3$ |
| VATEX | SigLIP2-G-OPT | 27.52 | 57.70 | 70.06 | 41.27 | 0.289 | 0.352 | $-1.7 \times 10^4$ | 0.660 | $-1.7 \times 10^4$ |
| | PECoreG | 18.90 | 42.42 | 54.72 | 30.42 | 0.379 | 0.480 | $-1.4 \times 10^4$ | 0.508 | $-1.4 \times 10^4$ |
| | V-SONAR | **40.75** | **68.63** | **78.88** | **53.59** | **0.427** | **0.558** | $-1.2 \times 10^4$ | **1.660** | $-8.3 \times 10^3$ |

Table 2: Zero-shot Retrieval performance on PE-VIDEO, DREAM-1K and VATEX. We report the Recall rate at 1/5/10 and MRR scores. We also report the analytical metrics for the embedding space, including 1) trace reflects overall variance, and 2) log determinant (logdet) approximates volume in the space. Best values for each columns are **bolded**.

similarity search. The metric XSIM++ includes hard negatives (Chen et al., 2023). We provide an ablation of the two SONAR versions for vision captioning tasks in Appendix E.

## 2.1 V-LCM

The Large Concept Model (LCM; LCM team et al. 2024) is a latent diffusion language model operating directly in the SONAR embedding space. It follows an auto-regressive paradigm, predicting the next sentence embedding conditioned on preceding clean embeddings. For the textual modality, all embeddings are encoded and decoded by the fixed SONAR encoder and decoder. To model the conditional distribution of the next embedding, LCM employs a diffusion objective: given a clean embedding $x^0 \in \mathbb{R}^d$, the forward process progressively perturbs it with Gaussian noise under a variance-preserving schedule (Karras et al., 2022):

$$q(x_t \mid x^0) = \mathcal{N}\left(x_t; \alpha_t x^0, \sigma_t^2 \mathbf{I}\right), \quad x_t = \alpha_t x^0 + \sigma_t \epsilon, \ \ \epsilon \sim \mathcal{N}(0, I), \tag{2}$$

where $(\alpha_t, \sigma_t)$ are determined by a monotonically decreasing log-SNR schedule $\lambda_t = \log(\alpha_t^2 / \sigma_t^2)$. The reverse process is parameterized by a denoiser $\mu_\theta(x_t, t, c)$, conditioned on the context embeddings $c$, with Gaussian transitions:

$$p_\theta(x_{t-1} \mid x_t, c) = \mathcal{N}\left(x_{t-1}; \mu_\theta(x_t, t, c), \sigma_t^2 \mathbf{I}\right). \tag{3}$$

Training minimizes a reconstruction loss on the original clean embedding:

$$\mathcal{L}(\theta) = \mathbb{E}_{t, x^0, \epsilon} \left\| x^0 - \mu_\theta(\alpha_t x^0 + \sigma_t \epsilon, t, c) \right\|_2. \tag{4}$$

We use the two-tower variant of LCM, which separates the contextualizer (encoding the preceding embeddings) from the denoiser (iteratively reconstructing the next embedding).

From the perspective of vision–language modeling, LCM represents a new paradigm that fuses information from visual and textual modalities within a modality-agnostic latent space prior, rather than discrete visual and textual tokens (Chameleon Team, 2024). This enables autoregressive generation to be performed entirely in the latent space. Building on this principle, we further introduce V-LCM, an extension of LCM trained through vision–language instruction fine-tuning to enhance its utility across a broad range of downstream vision-language tasks. In V-LCM, visual inputs (images and videos) are encoded into the SONAR latent space using V-SONAR, while textual instructions and prompts are encoded with SONAR. The resulting visual and textual embeddings are concatenated into a single sequence, which is then processed under the same latent diffusion framework as in LCM's original text-only training, predicting the next embedding in the sequence.

## 3 EXPERIMENTS

We first verify the effectiveness of aligning the vision encoder to SONAR, by evaluating text-video retrieval and captioning using V-SONAROmni aligned with the OMNISONAR text encoder, and

| Model | BLEU | R-1 | R-2 | R-L | BS-P | BS-R | BS-F |
|---|---|---|---|---|---|---|---|
| **PE-VIDEO** | | | | | | | |
| InternVL2.5-1B | 19.4 | 32.1 | 9.0 | 23.4 | 31.2 | 27.3 | 29.3 |
| InternVL2-1B | 24.1 | 35.8 | 10.7 | 25.5 | 30.8 | 32.1 | 31.5 |
| Qwen2-VL-2B-Instruct | 29.9 | 41.7 | 18.8 | 31.2 | 34.8 | 40.0 | 37.3 |
| Qwen2.5-VL-3B-Instruct | 30.0 | 41.3 | 16.1 | 28.9 | 30.2 | 38.6 | 34.4 |
| PLM-1B | 21.5 | 37.6 | 11.9 | 26.6 | 35.8 | 26.2 | 31.0 |
| PLM-3B | 21.1 | 37.5 | 11.7 | 26.4 | 36.6 | 26.1 | 31.3 |
| V-SONAR w/ OMNISONAR Decoder | **39.0** | **50.1** | **23.3** | **38.0** | **44.4** | **41.6** | **43.0** |
| **DREAM-1K** | | | | | | | |
| InternVL2.5-1B | 10.2 | 21.5 | 3.7 | 15.6 | **26.1** | 11.2 | 18.6 |
| InternVL2-1B | 14.6 | 25.0 | 4.3 | 17.2 | 23.8 | 15.2 | 19.5 |
| Qwen2-VL-2B-Instruct | 19.7 | 27.1 | 5.2 | 18.5 | 12.9 | 14.8 | 13.9 |
| Qwen2.5-VL-3B-Instruct | 16.1 | 23.9 | 4.4 | 15.9 | 1.6 | 15.6 | 8.6 |
| PLM-1B | 18.5 | 27.0 | 6.4 | 19.3 | 14.5 | 16.8 | 15.5 |
| PLM-3B | 19.6 | 28.6 | 6.7 | 20.4 | 19.9 | 18.1 | 19.0 |
| V-SONAR w/ OMNISONAR Decoder | **23.9** | **32.7** | **8.4** | **22.7** | 19.7 | **21.6** | 20.7 |
| **VATEX** | | | | | | | |
| InternVL2.5-1B | 41.5 | 23.3 | 4.4 | 19.2 | 37.6 | 45.2 | 40.2 |
| InternVL2-1B | **47.8** | **27.3** | **6.4** | **22.4** | **36.9** | 50.4 | **42.4** |
| Qwen2-VL-2B-Instruct | 32.1 | 19.8 | 6.0 | 16.4 | 18.7 | 46.3 | 30.8 |
| Qwen2.5-VL-3B-Instruct | 29.4 | 18.3 | 5.1 | 15.0 | 12.1 | 47.1 | 27.6 |
| PLM-1B | 33.4 | 21.8 | 5.7 | 19.1 | 15.1 | 48.1 | 29.6 |
| PLM-3B | 34.0 | 22.1 | 5.9 | 19.3 | 16.9 | 48.6 | 30.8 |
| V-SONAR w/ OMNISONAR Decoder | 26.7 | 17.2 | 5.0 | 14.8 | 13.6 | 42.8 | 26.5 |
| **VATEX-zh** | | | | | | | |
| InternVL2-1B-Instruct | 22.3 | 14.1 | 2.9 | 11.8 | 8.7 | 18.2 | 12.6 |
| InternVL2.5-1B-Instruct | **33.2** | 22.5 | 4.4 | 18.8 | 19.1 | 31.5 | 24.2 |
| V-SONAR w/ OMNISONAR Decoder | 30.6 | **32.1** | **8.53** | **26.9** | **23.2** | **48.5** | **33.7** |

Table 3: Video captioning performance across PE-VIDEO, DREAM-1K and VATEX (English and Chinese). Metrics include BLEU, ROUGE (R-1, R-2, R-L), and BERTScore (BS-P, BS-R, BS-F).

provide several ablations. We then switch to the zero-shot evaluation for LCM, and evaluation of V-LCM on M3IT (Li et al., 2023) which requires the use of V-SONAR1, as the LCM had been trained on SONAR1.

### 3.1 CONCEPT SPACE ALIGNMENT USING V-SONAROMNI

**Text-video Retrieval** We treat V-SONAR as a paired vision–text encoder and begin by evaluating its zero-shot performance on text-to-video retrieval, following the setup in Bolya et al. (2025). We compare V-SONAR against two strong baselines: the state-of-the-art SIGLIP2 vision encoder (Tschannen et al., 2025) and the PERCEPTION ENCODER (Bolya et al., 2025), from which V-SONAR is derived. Evaluations are conducted on three widely used video captioning benchmarks: PE-VIDEO (15K pairs of captioning data) (Bolya et al., 2025), VATEX (5K pairs of captioning data) (Wang et al., 2019), and DREAM-1K (1K pairs of captioning data) (Wang et al., 2024a), following the protocol in Cho et al. (2025). In addition to standard retrieval metrics such as Recall@1/5/10, we introduce three complementary measures to analyse embedding space properties: (1) Alignment Consistency (AC): the rank correlation between vision and text similarity scores, reflecting cross-modal alignment quality. (2) Trace: the trace of the covariance matrix of vision and text embeddings, indicating the spread of representations. (3) Log-determinant (logdet): the logarithm of the determinant of the covariance matrix, interpreted as the volume of the embedding ellipsoid.

Table 2 summarizes the results and embedding space statistics. On the three datasets, PE-VIDEO, DREAM-1K and VATEX, V-SONAR significantly outperforms SigLIP2, achieving improvements of 9.12, 1.8 and 13.23 points in Recall@1, respectively, demonstrating the effectiveness of our approach

for retrieval tasks. Compared to the original PERCEPTION ENCODER, V-SONAR significantly improves on PE-VIDEO and VATEX (9.12 and 21.85 score at Recall@1), though it loses 8.8 score at Recall@1 in DREAM-1K, indicating that our curriculum alignment strategy preserves strong retrieval capability. These results confirm that a vision encoder can be successfully aligned with a purely text-trained embedding space (SONAR) in a post-hoc manner. Finally, our embedding space analysis reveals that V-SONAR maintains a more expanded distribution. Moreover, by freezing the original SONAR space, V-SONAR achieves the largest textual embedding dispersion, as evidenced by the highest trace and logdet values among all compared models.

**Text-video Captioning** Different with the traditional vision encoder, such as SigLIP 2 or PERCEPTION ENCODER, aligning V-SONAR to SONAR embedding space allows us to leverage the SONAR decoder to directly verbalize the encoded vector of V-SONAR. Hence, we conduct the zero-shot evaluation on video captioning for V-SONAR, and compare it with few state-of-the-art vision-language models (VLMs) including InternVL-2/2.5 (Chen et al., 2024b), Qwen-VL 2/2.5 (Wang et al., 2024c; Bai et al., 2025) and Perception Language Models (Cho et al., 2025). We compare with VLMs at the scale between 1B to 3B for a fair comparison, as the SONAR decoder is at 1.5B and V-SONAR is at 1.9B. We evaluate the models with lexical metrics including BLEU and ROUGE scores, and semantic metrics including BERTScore-Precision/Recall/F1, following (Zhang et al., 2025).

We illustrate the results in Table 3. For detailed captioning benchmarks such as PE-VIDEO and DREAM-1K, we observe that V-SONAR paired with the SONAR decoder can achieve a state-of-the-art performance. In particular, V-SONAR improves the second best model, Qwen2.5-VL-3B-Instruct, by 9 points in BLEU. The only exception is VATEX where the captions are relatively short as one sentence, V-SONAR lags behind InternVL2; however, this is expected as we align V-SONAR with SONAR mostly with the detailed caption data. And we observe V-SONAR is still comparable with PLM and Qwen-VL series. We use VATEX-Chinese validation set for the multilingual evaluation, and we mostly compare V-SONAR with InternVL, as QwenVL is reported to leverage VATEX Chinese split during training (Wang et al., 2024c; Bai et al., 2025), and PLM-1/3B fail to support the fluent generation in Chinese. We find that in VATEX Chinese split, V-SONAR still outperforms the InternVL series, indicating the advantage in multilingual evaluation.

**V-SONAR 1 vs V-SONAROmni.** We present the comparison between SONAR (Duquenne et al., 2023) and OMNISONAR in Figure 2. We report both SONAR space's oracle performance (we encode the reference caption with SONAR encoder, and decoded with SONAR decoder). And we report the zero-shot performance with V-SONAR (we encode the video with V-SONAR and decode with SONAR decoder). SONAR oracle serves as an estimation of the upper-bound performance that V-SONAR can achieve for leveraging SONAR decoder.

We observe that both SONAR versions have a strong oracle performance, indicating SONAR's encoding and decoding from textual space into its representation space is quite lossless. Specifically, in PE-Video, VATEX and DREAM-1K, OMNISONAR can achieve BLEU scores of 81, 96 and 70. Comparing the zero-shot performance for SONAR and OMNISONAR, we see SONAR1 is worse by a considerable margin. We hypothesize that SONAR is harder to align since its space is reported to be collapsed. Our analysis for SONAR and OMNISONAR also supports this observation: in PVD, SONAR and OMNISONAR have the embeddings norm at 0.264 and 1.69, and covariance trace at 0.049 and 1.83, respectively. The comparison in retrieval tasks are in Appendix E.

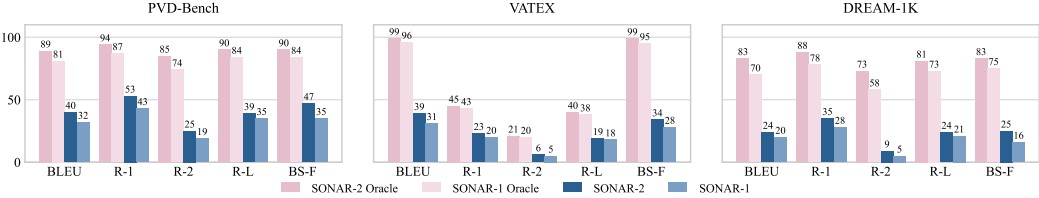

Figure 2: Comparison for V-SONAR trained with SONAR version 1 and 2 (OMNISONAR) embedding space on three captioning datasets.

| | | MSE | Cos. Sim. | BLEU | R-1 | R-2 | R-L | BS-P | BS-R | BS |
|---|---|---|---|---|---|---|---|---|---|---|
| **Architecture** | Linear Proj. | $1.45 \times 10^{-3}$ | 0.694 | 38.0 | 49.7 | 21.6 | 36.7 | 47.2 | 40.1 | 43.7 |
| | Full PE | $1.54 \times 10^{-3}$ | 0.672 | 37.1 | 48.5 | 21.3 | 36.5 | 46.9 | 38.8 | 42.9 |
| | + Async. LR | $1.43 \times 10^{-3}$ | 0.700 | 39.7 | 51.3 | 23.3 | 37.7 | 48.1 | 42.1 | 45.1 |
| | + Norm. Init. | $1.39 \times 10^{-3}$ | 0.708 | 39.8 | 51.8 | 24.0 | 38.5 | 49.4 | 42.2 | 45.8 |
| | + Attn. Pooling | $1.39 \times 10^{-3}$ | 0.708 | 39.8 | 51.9 | 24.0 | 38.5 | 49.7 | 42.4 | 46.0 |
| | + Temporal Attn. | $1.39 \times 10^{-3}$ | 0.708 | 39.8 | 51.9 | 24.0 | 38.5 | 49.7 | 42.4 | 46.1 |
| **Pipeline** | Full Pipeline | $1.36 \times 10^{-3}$ | 0.716 | 40.1 | 52.6 | 24.9 | 39.2 | 50.8 | 43.2 | 47.0 |
| | *w/o* SV | $1.39 \times 10^{-3}$ | 0.710 | 39.6 | 51.9 | 24.1 | 38.6 | 50.0 | 42.4 | 46.2 |
| | *w/o* IC & SV | $1.39 \times 10^{-3}$ | 0.708 | 39.8 | 51.9 | 24.0 | 38.5 | 49.7 | 42.4 | 46.1 |

Table 4: Ablation study in model architecture and the three-stage training pipeline. SV: our second stage curriculum with the synthetic video captioning data. IC: our first stage curriculum with image captioning data.

## 3.2 ABLATION STUDY

We conduct an ablation study for model architecture design, and our proposed training pipeline on the PE-VIDEO test set (Table 4).

**Model Architecture** We ablate architectural choices for the projector network. As a baseline, we evaluate linear projection (Linear Proj.), where the PERCEPTION ENCODER is frozen and only a linear layer is trained, and full-model fine-tuning (Full PE), where the encoder is updated jointly. Linear projection performs better, indicating that the encoder's contrastive pre-training already yields strong semantic alignment, while full fine-tuning is hindered by unstable gradients from the randomly initialized projector. To mitigate this, we adopt strategies that incrementally improve downstream performance, including asynchronous learning rates for the projector and encoder, initialization trick, attention-based aggregation strategy for video frames' features, and temporal attention layer.

**Data Mixture** We then ablate the second stage synthetic video captioning and first-stage image captioning curriculum on our pipeline. We observe that both stages contribute positively to the downstream performance in captioning performance on PE-Video. Removing the 12M image captioning pairs and 2M video captioning pairs reduce the BLEU with 0.3 and 0.2, respectively.

## 3.3 ZERO-SHOT PROCESSING V-SONAR EMBEDDINGS BY LARGE CONCEPT MODEL

Since the LCM (LCM team et al., 2024) operates directly on SONAR1, it should seamlessly transfer its ability and understand the visual concepts in V-SONAR aligned with SONAR1. We examine V-SONAR with LCM gradually from single to multiple vision concept understanding tasks, where the LCM accepts the instruction encoded by SONAR, with the vision embeddings from V-SONAR, and predicts the target embedding. Note that in both experiments, we do not fine-tune the LCM, with neither any video data nor captioning data. Thus, the LCM is only trained in English textual corpus including its pre-training and instruction fine-tuning as in LCM team et al. (2024). We compare LCM's performance with VLMs at 7/8-B scale for the InternVL series (Chen et al., 2024b), Qwen-VL (Bai et al., 2025; Team, 2024) and PLM (Cho et al., 2025).

**Single Vision Concept Understanding: Video Captioning** We report the results of LCM on video captioning in Table 5. Compared to the strongest baseline, the zero-shot LCM lags behind by 1.15/4.44/4.76 BLEU scores on PE-VIDEO, DREAM-1K and VATEX, respectively. Among the models, PLM-8B delivers the strongest overall performance. The relatively narrow performance gap between LCM and competitive VLMs suggests that the LCM is able to understand the single vision concept, despite never being trained with video data.

**Multiple Vision Concept Understanding: Long Video Summarization** We next evaluate LCM in a setting requiring understanding multiple visual embeddings. Long videos are uniformly segmented

| | | Video Captioning / Summarization | | | | | | | | M3IT Image | | | | M3IT Video | | |
|---|---|---|---|---|---|---|---|---|---|---|---|---|---|---|---|---|---|
| | | PE-Video | | DREAM-1K | | VATEX | | VIDEOXUM | | COCO | VIQUAE | VisualMRC | ScienceQA | ActivNetQA | MSRVTT-QA | IVQA |
| | | R-L | BS | R-L | BS | R-L | BS | R-L | BS | R-L | R-L | R-L | Acc. | R-L | R-L | R-L |
| **InterVL2** | 1B | 25.5 | 31.5 | 17.2 | 19.5 | 22.4 | 42.4 | 15.3 | 17.7 | 12.6 | 24.0 | 30.6 | 53.9 | 40.6 | 27.6 | 39.5 |
| | 4B | 15.0 | 18.4 | 12.2 | 14.7 | 19.2 | 42.4 | 15.6 | 17.4 | 17.5 | 20.0 | 35.3 | 89.6 | 27.5 | 24.5 | 31.8 |
| | 8B | 18.6 | 23.4 | 16.4 | 19.4 | 16.2 | 42.4 | 29.1 | 26.1 | 21.0 | 21.6 | 42.9 | 87.2 | 29.7 | 27.1 | 38.4 |
| **InternVL-2.5** | 1B | 23.4 | 29.3 | 15.6 | 18.6 | 19.2 | 40.3 | 17.1 | 23.2 | 13.2 | 10.8 | 27.3 | 69.0 | 16.6 | 11.7 | 19.3 |
| | 4B | 14.6 | 18.0 | 13.3 | 14.8 | 17.3 | 36.2 | 18.1 | 23.0 | 15.1 | 23.1 | 45.2 | 86.4 | 26.8 | 21.8 | 24.9 |
| | 8B | 21.4 | 26.0 | 17.0 | 17.0 | 20.6 | 42.4 | 24.9 | 20.5 | 16.8 | 17.3 | 42.8 | **93.1** | 20.9 | 16.9 | 22.7 |
| **Qwen2-VL** | 2B | 31.2 | **37.3** | 18.5 | 13.9 | 16.4 | 30.8 | 23.6 | 29.8 | 24.9 | **50.2** | 56.1 | 54.5 | 53.7 | 39.6 | 49.4 |
| | 7B | 26.9 | 32.6 | 19.8 | 18.1 | 28.5 | **51.6** | 26.0 | 32.4 | 23.7 | 49.7 | **57.4** | 70.4 | 41.9 | 22.7 | 39.1 |
| **Qwen2.5-VL** | 3B | **28.9** | 34.4 | 15.9 | 8.6 | 15.0 | 27.6 | 26.0 | 32.9 | 25.1 | 48.3 | 55.7 | 55.0 | 52.1 | 41.6 | 48.5 |
| | 7B | 22.2 | 25.9 | 15.7 | 10.5 | 27.5 | 50.8 | 24.1 | 28.9 | 18.5 | 34.5 | 45.0 | 61.6 | 46.0 | 41.4 | 54.2 |
| **Percep. LM** | 1B | 26.6 | 31.0 | 19.3 | 15.5 | 19.1 | 29.6 | 21.8 | 33.2 | 27.5 | 30.8 | 45.5 | 73.6 | 27.8 | 14.5 | 39.1 |
| | 3B | 26.4 | 31.3 | 20.4 | 19.0 | 19.3 | 30.8 | **27.0** | **36.4** | 34.3 | 23.7 | 51.1 | 89.8 | 28.0 | 19.4 | 26.1 |
| | 8B | 27.4 | 31.9 | **20.8** | **19.7** | 19.0 | 30.8 | 26.2 | 33.7 | 36.3 | 31.0 | 50.0 | 87.7 | 40.5 | 25.3 | 41.4 |
| **LCM** | LCM | 25.5 | 27.9 | 18.5 | 16.6 | 23.8 | 30.8 | 21.5 | 22.1 | 18.0 | 34.3 | 33.5 | 44.7 | 51.7 | 36.0 | 48.9 |
| | v-LCM | 27.4 | 30.0 | 19.8 | 19.2 | **28.8** | 48.7 | 20.6 | 25.3 | **38.8** | 39.4 | 34.1 | 76.2 | **63.6** | **48.7** | **63.9** |

Table 5: Main results on vision-language tasks in M3IT and the previous video benchmarks (PE-VIDEO, DREAM-1K, VATEX, VideoXum).

into snippets with 8 frames per each, with each snippet encoded by V-SONAR as a separate video embedding. Since the LCM shows strong performance in document summarization (LCM team et al., 2024) for multiple SONAR embeddings; we hypothesize that it should be capable of performing zero-shot summarization over sets of video embeddings from V-SONAR. For this evaluation, we use VIDEOXUM (Lin et al., 2023), which contains videos of one to five minutes, uniformly split into snippets of 8 frames each.

We report the VIDEOXUM results in Table 5. Again, PLM achieves the strongest performance among competitive VLMs at the same scale. V-SONAR + LCM achieves 22.1 score at BertScore-F1, trailing the best-performing PLM-8B (33.7) but is slightly higher than InternVL-2.5-8B at 20.5. These findings indicate that, even without exposure to any video data during training, LCM demonstrate non-trivial understanding of multiple V-SONAR embeddings for long videos.

**Reasoning in V-SONAR** We then investigate whether LCM truly leverages the latent representations in V-SONAR for multimodal reasoning. To test this, we compare two settings: (1) encoding video clips directly into V-SONAR embeddings, which are then fed into LCM for zero-shot summarization; and (2) decoding video embeddings into captions using the SONAR decoder, re-encoding them with SONAR, and providing these SONAR embeddings to LCM. Our hypothesis is that V-SONAR embeddings retain richer visual features than their textual equivalents in SONAR, and thus should yield stronger performance if LCM relies on visual representations.

We group videos into short (<90s), mid-length (90–150s), and long (>150s) categories, and report ROUGE-L scores in Figure 3. Across all categories, LCM with V-SONAR consistently outperforms it with SONAR. Notably, while SONAR performance declines with increasing video length,

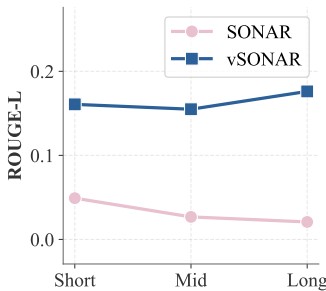

Figure 3: Operating in V-SONAR space, the LCM performs better than only accepting the textual SONAR inputs. We compare LCM-7B-IFT in VIDEOXUM with ROUGE-L scores across short, mid, and long categories of video inputs.

V-SONAR remains stable, highlighting its robustness. These results support our hypothesis that the LCM reasons directly in the visual embedding space provided by V-SONAR containing richer visual information than SONAR representations of textual input.

**v-LCM Reasoning with Visual Details.** To assess whether our alignment from PERCEPTION ENCODER to SONAR preserves the visual details critical for downstream reasoning, we further evaluate visual commonsense reasoning (VCR) performance on M3IT (Table 6). In this benchmark, models must reason about objects using commonsense knowledge while interpreting their associated bounding boxes. Compared to semantic-level alignment alone, VCR demands substantially richer visual grounding and layout awareness (see Figure 21). We report token-level F1 scores against the ground truth rationales, alongside semantic similarity to the reference explanations. The strong performance of v-LCM indicates that, despite being trained solely on semantic-

| VCR | F1 | Sim. |
|---|---|---|
| LCM | 0.385 | 0.258 |
| **vLCM** | **0.671** | **0.529** |
| PLM-8B | 0.441 | 0.432 |
| Qwen-2.5-7B | 0.275 | 0.402 |
| Qwen-2-7B | 0.502 | 0.513 |
| InterVL2.5-8B | 0.155 | 0.158 |
| InterVL2-8B | 0.325 | 0.340 |

Table 6: VCR results comparing token-level F1 and semantic similarity.

level captions, v-LCM effectively leverage layout grounding and spatial relationship preserved by v-SONAR in reasoning tasks.

### 3.4 v-LCM

**In-task Performance** We next evaluate v-LCM, which is supervised fine-tuned on M3IT (Li et al., 2023) to better capture and reason over visual concepts. We mainly rely on M3IT (Li et al., 2023) as it supports a variety of tasks, and the wide coverage for up to 80 languages ranging from high- to low-resource languages. The evaluation covers 7 datasets spanning 5 tasks defined in M3IT: (1) image captioning (COCO), (2) visual QA (VIQUAE), (3) document image QA (VisualMRC), (4) video captioning (MSRVTT), and (5) question answering (IVQA, MSRVTT-QA, ActivityNetQA). In addition, we report v-LCM's performance on the video captioning and long video summarization benchmarks introduced in the previous section. Following the evaluation protocol in Li et al. (2023), we use ROUGE-L for generative tasks (e.g., captioning and open-ended QA) and accuracy for multiple-choice QA.

Table 5 compares LCM with strong open-source vision–language models. We observe that v-LCM substantially outperforms the zero-shot LCM across most benchmarks. For example, v-LCM achieves 63.9 R-L on IVQA and 63.6 R-L on ActivityNetQA, surpassing 48.9 and 51.7 for LCM, representing clear gains from training with vision instruction-tuning data. While v-LCM lags behind the best-performing models on some benchmarks such as VisualMRC, VIQUAE, and ScienceQA, it achieves state-of-the-art results on video question answering tasks, including IVQA, ActivityNetQA, and MSRVTT-QA. Meanwhile, performance on our previous video captioning and summarization datasets remains competitive: v-LCM attains 27.4 R-L on PE-Video and 19.8 R-L on DREAM-1K, trailing the best model by only 1.5 and 1 ROUGE scores, highlighting the generalization of LCM to unseen datasets during training.

**Multilinguality** We further conduct a multilingual evaluation of v-LCM on the M3IT benchmark across 62 languages,[2] leveraging the fact that v-LCM operates entirely in the latent space of SONAR and v-SONAR, and can therefore decode outputs to any languages supported by SONAR. Evaluation spans five various tasks including image classification (ImageNet), image question answering (VQA-V2, OKVQA), video question answering (MSRVTT-QA), video captioning (MSRVTT) and narrative generation (VIST), covering a spectrum from high-resource languages (e.g., Chinese), mid-resource languages (e.g., Japanese) to low-resource languages (e.g., Javanese). We use the ROUGE-L implementation from (Shohan et al., 2024) for the multilingual evaluation.

As shown in Figure 4, v-LCM consistently outperforms Qwen2.5-VL-7B and PLM-8B across 61 of 62 languages, with Dutch being the only exception. While improvements in some high-resource languages are modest (e.g. French), the gains become substantial in mid- and low-resource settings, including Burmese, Tajik and Telugu. Notably, for languages such as Urdu, modern Arabic and Tamil, which is unsupported by PLM-8B based on LLaMA-3.2 (Touvron et al., 2023; Dubey et al., 2024), v-LCM successfully generates meaningful outputs, whereas competing models fail entirely.

---

[2]The intersection of all languages supported by SONAR, M3IT, and multilingual ROUGE.

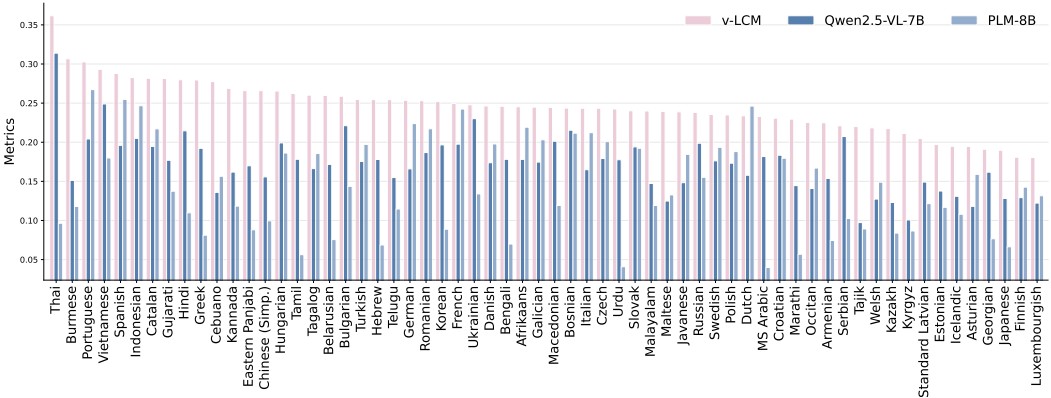

Figure 4: Performance in 62 languages for v-LCM, Qwen2.5-VL-7B and PLM-8B on M3IT testing set for MSRVTT, MSRVTT-QA, ImageNet, VQA-V2, VIST and OKVQA. We report the ROUGE-L scores averaged from all datasets. Detailed results for each dataset can be found in the Appendix F.

## 4 RELATED WORKS

A central paradigm in multimodal learning is to align vision and language representations into a shared embedding space. Early approaches such as CLIP (Radford et al., 2021) and ALIGN (Jia et al., 2021) established large-scale contrastive learning between paired images and captions, enabling zero-shot transfer to downstream tasks. Subsequent works extended this idea to video–language pretraining (Lei et al., 2021; Xu et al., 2021; Wang et al., 2022). More recent efforts focus on aligning pretrained encoders into an unified space: Perception Encoder (Bolya et al., 2025) projects diverse perceptual modalities into a shared latent space, while scaling data and architectures in models, such as Florence (Yuan et al., 2021) and SigLIP2 (Tschannen et al., 2025), further improve alignment quality. Recent work also shows that using large language models as text encoders enhances vision–language alignment (Stone et al., 2025), and post-hoc alignment strategies have been proposed as lightweight alternatives to joint training (Brokowski et al., 2025; Yang et al., 2025).

Parallel advances in multilingual text embedding models, such as LASER (Artetxe & Schwenk, 2019; Heffernan et al., 2022), LABSE (Feng et al., 2020), and SONAR (Duquenne et al., 2023; Omnilingual Embeddings Team et al., 2026), demonstrate the effectiveness of language-agnostic embedding spaces across hundreds of languages. Modular approaches have further explored language-specialized components to reduce interference in universal embedding spaces (Huang et al., 2024). These universal text embeddings provide an attractive target for aligning vision encoders, as they inherit cross-lingual generalization without requiring multimodal data in every language. Prior work has explored similar strategies in speech-to-text alignment (Chung et al., 2018; Duquenne et al., 2021a; Laperrière et al., 2024; Du et al., 2024), but large-scale alignment of visual embeddings into such universal text spaces remains underexplored.

## 5 CONCLUSION

We introduce V-SONAR by extending the SONAR embedding space with the image and video modality. To the best of our knowledge, this makes SONAR the most universal embedding space covering four modalities (text, speech, image and video) and up to 200 languages. We propose a three-stage training approach to map a pooled representation based on the PERCEPTION ENCODER to the semantic SONAR representation. We achieve very competitive results for text-to-video retrieval and video captioning. The Large Concept Model (LCM; LCM team et al. 2024) is a recent approach to perform reasoning at a higher semantic conceptual level, namely SONAR. Encoded by V-SONAR, we show that the LCM can zero-shot process image or video embeddings without the need of training data in these modalities. We further introduce v-LCM with the multimodal instruction fine-tuning, which matches state-of-the-art VLMs, while significantly outperforming them across 61 rich- to low-resource languages.

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

## A    LLMs Usage Declaration

We declare that the large language model (LLM) was only used to assist in minor tasks, including revising the manuscript for grammatical correctness, improving phrasing, and performing small technical implementations such as debugging code snippets. All scientific ideas, results, analyses, and conclusions presented in this paper are entirely the work of the authors.

## B    Contrastive Loss for Aligning Perception Encoder and Sonar

We have also explored the use of a contrastive loss in addition to the MSE loss for aligning the Perception Encoder to Sonar. Specifically, given a mini-batch of $B$ paired samples $\{(V_i, T_i)\}_{i=1}^{B}$, we aim to not only minimize the distance between matched pairs $(f_\theta(V_i), g(T_i))$ but also push apart mismatched pairs. We define the contrastive loss as:

$$\mathcal{L}_{\text{con}} = -\frac{1}{B} \sum_{i=1}^{B} \log \frac{\exp\left(\text{sim}(f_\theta(V_i), g(T_i))/\tau\right)}{\sum_{j=1}^{B} \exp\left(\text{sim}(f_\theta(V_i), g(T_j))/\tau\right)}, \quad (5)$$

where $\text{sim}(\cdot, \cdot)$ denotes cosine similarity and $\tau$ is a temperature parameter. The final loss is then a weighted combination of the MSE alignment loss and the contrastive loss:

$$\mathcal{L} = \mathcal{L}_{\text{align}} + \lambda \mathcal{L}_{\text{con}}, \quad (6)$$

where $\lambda$ controls the strength of the contrastive term. However, in our preliminary experiments, adding the contrastive component did not yield a significant improvement over the MSE-only objective (Table 7) in captioning performance, while it leads to gains in retrieval performance. However, since our downstream usage of v-Sonar for v-LCM is closer to generation task, we choose the MSE-only loss as the final loss.

| | Captioning | | | Retrieval | |
|---|---|---|---|---|---|
| | Bleu | R-L | BS-F1 | R@1 | MRR |
| MSE-only | **38.9** | **37.8** | **44.9** | 49.0 | 60.3 |
| MSE + Contrastive | 38.6 | 37.5 | 44.5 | **52.4** | **63.7** |

Table 7: Ablation study on using only MSE loss vs. adding a contrastive loss. Results are reported on the PE-Video benchmark for captioning and retrieval with a single MLP layer as the connector in v-Sonar.

To further explain the trade-off between retrieval and captioning performance in Table 7, we hypotheses that contrastive training can move vision vectors off the exact SONAR manifold. It only enforces relative ordering via cosine margins, and produces embeddings whose norms or local covariance differ from those the SONAR decoder were trained on. That small manifold shift can degrade generative reconstruction even if retrieval improves.

We analyze the statistics of v-Sonar embeddings in Table 8. It is a clear observation that v-Sonar trained with the contrastive loss has a more separate embeddings, evidenced by its higher norm, trace and volume. It shows that contrastive loss pushes the vSONAR embeddings to be a more expanded distribution compared to the MSE-only (higher values in Norm, Covariance trace, and Volume). However, their poor alignment consistencies in both MSE and cosine similarity, where we calculate the correlation between vision and text similarity ranking for each sample, suggests contrastive loss breaks the local covariance structure and may break the alignment with SONAR manifold.

## C    Dataset Statistics

Table 9 summarizes the datasets used in our three-stage training pipeline for alignment. The PLM-Image datasets (SA1B and OpenImages) provide large-scale image–caption pairs, which are particularly valuable for improving grounding and linguistic richness. The PLM-Video-Auto-YT1B dataset contributes video–text pairs with an average duration of 22.75 seconds, enabling the model to capture

|  | V. Norm | V. Trace | V. Volume | AC (Cosine) | AC (MSE) |
|---|---|---|---|---|---|
| MSE | 1.22 | 0.48 | -10007.16 | **0.41** | **0.32** |
| MSE+Contrastive | **1.30** | **1.74** | **-8107.94** | 0.31 | 0.13 |

Table 8: Additional statistics for V-SONAR representations on PE-Video. We report the norm (V.Norm), trace (V.Trace), Volume (V.Volume), alignment consistency measured in Cosine Similarity (AC Cosine) and MSE (AC MSE) for video embeddings predicted by V-SONAR.

temporal dynamics in multimodal content. Finally, the PE-Video dataset provides carefully curated human-annotated video–caption pairs with moderate length, serving as a higher-quality supervision signal in later stages. Together, these datasets balance scale and quality, ensuring both broad coverage and precise alignment across modalities.

| Dataset | #Samples | Duration (s) | Caption Length (sent.) | Caption Length (words) |
|---|---|---|---|---|
| PLM-Image-Auto-SA1B | 7.99M | – | 10.7 | 181.8 |
| PLM-Image-Auto-OpenImages | 1.37M | – | 7.9 | 132.1 |
| PLM-Video-Auto-YT1B | 2.14M | 22.8 | 2.3 | 95.5 |
| PE-Video | 118K | 16.7 | 4.4 | 51.4 |

Table 9: Statistics of the datasets used in 3-stage training for alignment. We report the number of samples, average video duration (if applicable), and average caption length in sentences and words.

## D    IMPLEMENTATIONS

**V-SONAR Architecture**   We build our model on top of the Perception Encoder Vision Transformer backbone `PE-Core-G14-448`[3]. The encoder processes RGB images at a resolution of 448×448 pixels, splitting them into 14×14 patches and yielding 1024 patches per frame. The vision tower consists of 50 transformer layers, each with a hidden width of 1024, 16 attention heads, and a 4096-dimensional feed-forward network, resulting in approximately 1.9B parameters. For video inputs, we uniformly sample 8 frames and extract frame-level embeddings of 1536 dimensions from the encoder, which are subsequently projected into a 1024-dimensional SONAR embedding space. To bridge perception features with the target space, we attach a lightweight connector, where weights are optionally initialized from a Gaussian distribution ($\mu = 0, \sigma = 1e - 5$) with zero biases for stability. To capture temporal dynamics, the connector augments encoder outputs with sinusoidal positional encodings and applies a temporal multi-head self-attention module (8 heads, dropout 0.1) across frames, combined with residual connections. The resulting sequence is aggregated using attention-based pooling, where a learnable CLS token attends over the frame embeddings via an 8-head attention module, though we also evaluate mean and max pooling variants. The final pooled representation (1536 dimensions) is then mapped to the 1024-dimensional SONAR space by a linear MLP layer.

**V-SONAR Training Details**   To stabilize training, the projector is initialized from a zero-mean Gaussian distribution with a small variance ($1e$–5), which mitigates gradient explosion when mapping from the high-dimensional PERCEPTION ENCODER features to the target embedding space. We employ a two-phase training recipe: in the first 2,000 steps, PERCEPTION ENCODER is frozen while only the projector is optimized, allowing the projector to adapt without perturbing the pre-trained encoder. Subsequently, both the projector and PERCEPTION ENCODER are jointly optimized, using asynchronous learning rates: a higher rate ($1e$–4) for the projector to enable rapid adaptation, and a lower rate ($1e$–5) for PERCEPTION ENCODER to preserve pre-trained knowledge.

We train the model using a three-stage curriculum. Stage 1 (image captioning) runs for 15 epochs with a batch size of 512, a base learning rate of $1 \times 10^{-5}$, and a connector learning rate of $1 \times 10^{-4}$, with 4000 warmup steps applied to the connector. Stage 2 (synthetic video captioning data) runs for 10 epochs with an effective batch size of 128, a learning rate of $1 \times 10^{-5}$, and a connector learning

---

[3]https://huggingface.co/facebook/PE-Core-G14-448

rate of $1 \times 10^{-4}$ with 2000 warmup steps. Stage 3 (manually verified video captioning data) adopts the same settings as Stage 2. Across all stages, we optimize with AdamW, cosine learning rate decay, and a 500-step linear warmup schedule. We fix the random seed to 42 and evaluate on 2000 validation samples per stage. Training is distributed with Fully Sharded Data Parallel (FSDP) across 64 Nvidia A100-80G GPUs using `bfloat16` precision, gradient accumulation for memory efficiency, and early stopping with a patience of 3 epochs, checkpointing the best validation model.

**V-LCM Training Details**  For training the V-LCM, we adopt the LCM two-tower architecture (LCM team et al., 2024) with the diffusion-based next-sentence fine-tuning objective. The optimizer is AdamW with $\epsilon = 10^{-6}$, weight decay of 0.01, gradient clipping at 25.0, and learning rate $3 \times 10^{-5}$ scheduled with cosine decay, warmed up over the first 300 steps, and annealed to a final learning rate of $10^{-6}$. Training is run for a maximum of 10,000 steps with batch sizes dynamically determined up to 7168 latent embeddings, using gradient accumulation set to 1. Checkpoints are saved every 1000 steps, and we select the best performance according to the validation performance. The criterion incorporates a conditional guidance probability of 0.15 as used in LCM, and loss is reduced with a summation loss function. Training uses Fully Sharded Data Parallel (FSDP) with bf16 precision for efficiency. Data loading is set to uniformly sampled from all training set in M3IT, length-ordered batching without packing. Experiments are conducted on 1 node with 8 A100 GPUs (80GB).

## E  SONAR VS OMNISONAR

We also report the comparison in the retrieval performance in Table 10. While OMNISONAR generally holds an advantage, SONAR1 remains highly competitive (e.g., R@1 of 64.9 on PVD-Bench compared to PECoreG).

|  |  | **R@1** | **R@5** | **R@10** | **MRR** |
|---|---|---|---|---|---|
| **PVD-Bench** | SONAR1 | 0.649 | 0.843 | 0.895 | 0.737 |
| | OMNISONAR | 0.730 | 0.898 | 0.938 | 0.805 |
| **DREAM-1K** | SONAR1 | 0.536 | 0.759 | 0.838 | 0.638 |
| | OMNISONAR | 0.633 | 0.841 | 0.890 | 0.725 |
| **VATEX** | SONAR1 | 0.119 | 0.263 | 0.350 | 0.195 |
| | OMNISONAR | 0.408 | 0.686 | 0.789 | 0.536 |

Table 10: Comparison for V-SONAR trained with SONAR version 1 and 2 embedding space on three text-to-video retrieval datasets.

## F  DETAILED MULTILINGUAL EVALUATION

We present the results of our multilingual evaluation across all supported datasets in this section. Specifically, we test all languages covered by SONAR and M3IT, reporting ROUGE scores as the primary metric. Figure 10 shows the results for the image captioning task on ImageNet, while Figure 5 and Figure 6 report results for video captioning and video QA on MSR-VTT, respectively. Figure 7 presents results on OKVQA, Figure 8 illustrates performance on story generation in VIST, and Figure 9 shows results for image QA on VQA-v2. With the exception of ImageNet, likely due to its widespread use and extensive coverage in existing VLMs, our model consistently outperforms baselines across all tested languages, with the only exception being Thai in VQA-v2.

## G  VISUALIZATION FOR V-SONAR'S LATENT SPACE

To qualitatively assess the effectiveness of our alignment, we visualize the latent spaces of video and SONAR embeddings before and after each stage of aligning PERCEPTION ENCODER to SONAR using t-SNE (Figure 11). After each stage of alignment, we observe a better clustering structure where video embeddings and their corresponding SONAR embeddings lie in the closer proximity, indicating that the alignment successfully reduces modality gaps in the latent space and supports a shared semantic representation across modalities, thereby validating the alignment strategy.

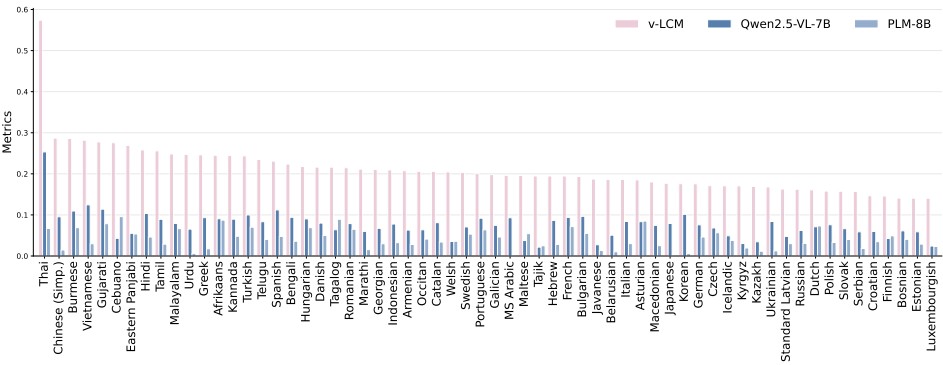

Figure 5: M3IT evaluation on 61 languages for MSRVTT.

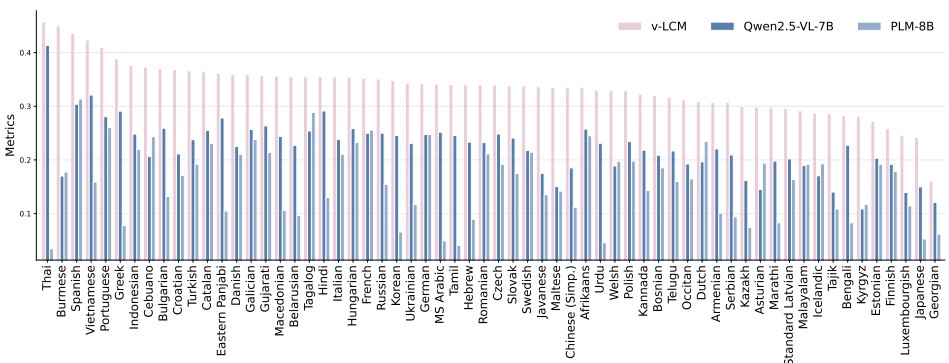

Figure 6: M3IT evaluation on 61 languages for MSRVTT-QA.

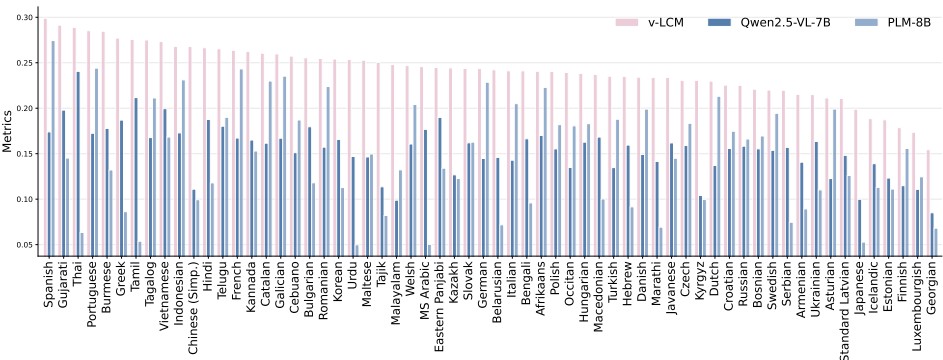

Figure 7: M3IT evaluation on 61 languages for OKVQA.

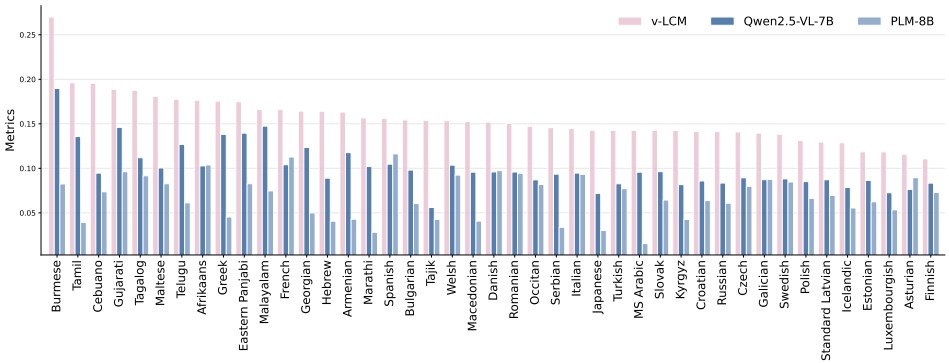

Figure 8: M3IT evaluation on 61 languages for VIST.

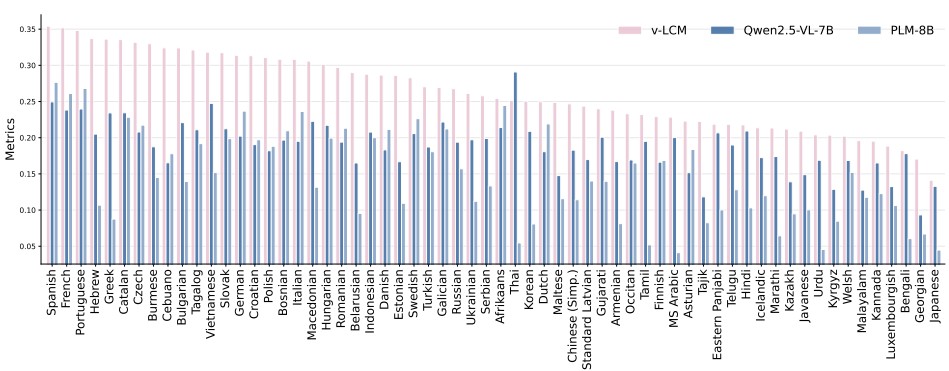

Figure 9: M3IT evaluation on 61 languages for VQA-V2.

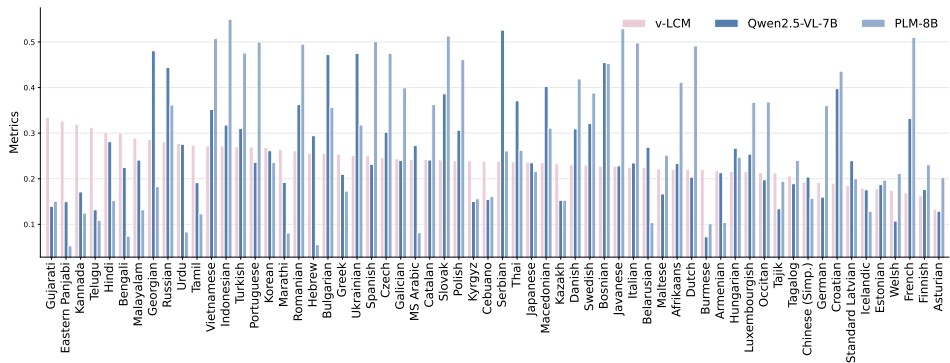

Figure 10: M3IT evaluation on 61 languages for image captioning.

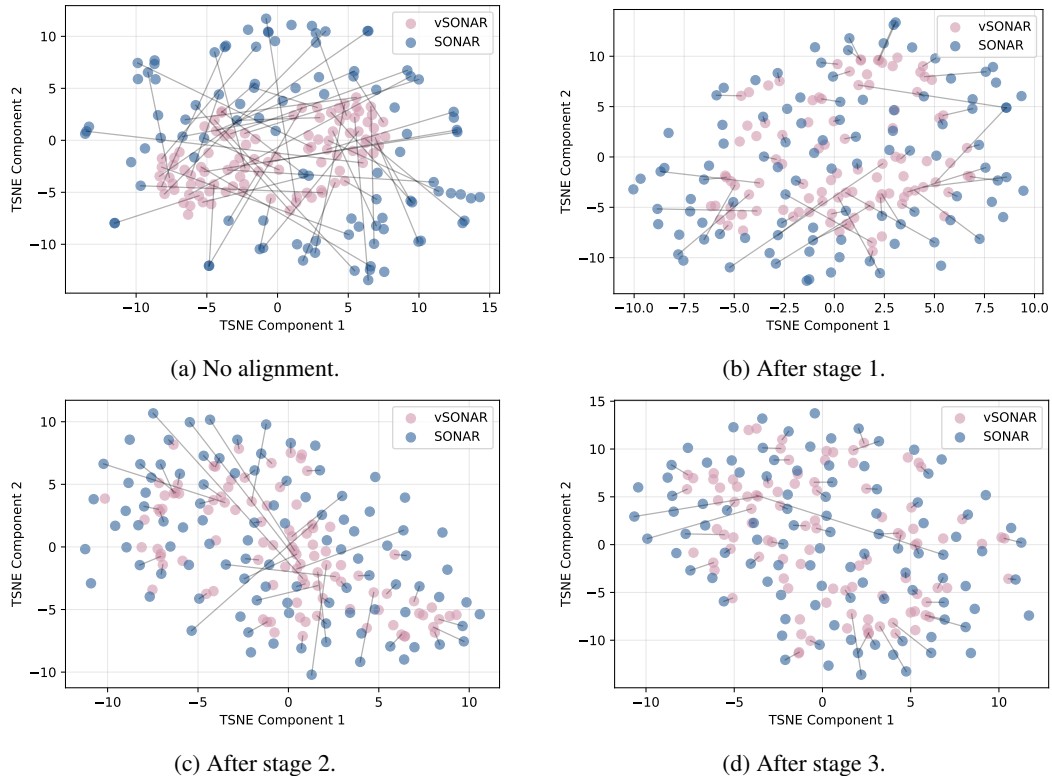

Figure 11: Visualization with t-SNE for SONAR and V-SONAR embeddings after each stage of curriculum. V-SONAR encodes the video, and SONAR encodes the caption. We randomly sample 200 samples from PE-Video's testing set for t-SNE, and explicitly plot the lines for connecting the paired video and caption for 50 samples.

## H   ANALYSIS IN CROSS-MODAL DRIFT FOR V-SONAR AND V-LCM

In this section, we investigate the potential happening of the cross-modal drift in V-SONAR and V-LCM's decoding process. We aim to answer the question whether the generated language by SONAR decoder and V-LCM faithfully represents the same semantic region of the embedding space in V-SONAR. We conduct three analysis on PE-Video testing set to empirically prove that V-SONAR and V-LCM suffered a minimum semantic drift during decoding when they handle embeddings from various modalities.

|  | Semantic Sim. | | Round-trip Retrieval | | | |
|---|---|---|---|---|---|---|
|  | Cosine | Dist. | R@1 | R@5 | R@10 | MRR |
| Groundtruth | 0.666 | 0.197 | 87.00% | 95.90% | 97.10% | 0.9084 |
| SONAR Decoder | 0.689 | 0.175 | 82.50% | 97.00% | 98.70% | 0.8883 |
| V-LCM | 0.562 | 0.219 | 82.30% | 96.70% | 97.90% | 0.8867 |

Table 11: Left: semantic similarities and distances between video embedding and the ground-truth captions, generated captions from SONAR Decoder and V-LCM. Right: results for the round-trip retrieval ablation study where we use the ground-truth captions, SONAR Decoder and V-LCM's generated captions to retrieve the videos.

**Embedding-Level Semantic Fidelity.**   For each video embedding $v$, we first compare its similarity and distance to: (i) ground-truth caption embedding $t_{gt}$, (ii) SONAR Decoder caption embedding ($t_{sonar}$, and (iii) LCM caption embedding $t_{lcm}$. This analysis provides a direct comparison for the generated captions from SONAR decoder and V-LCM with the ground-truth captions.

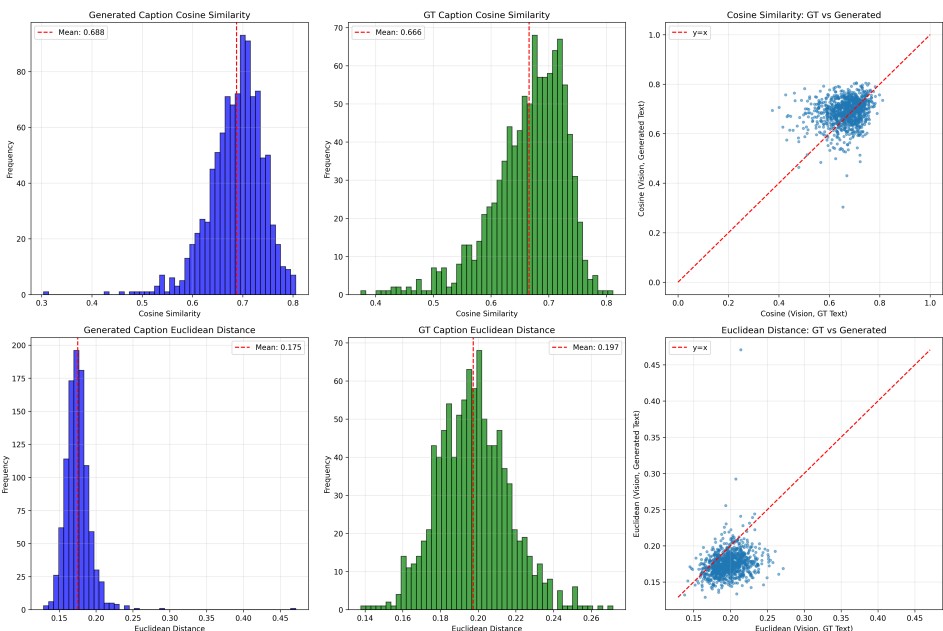

Figure 12: vSONAR's visualization for cross-modal semantic drift.

As indicated in Table 11, we find that SONAR-decoded captions show nearly identical cosine similarity or distance compared to the groundtruth, indicating negligible cross-modal drift. vLCM captions show a slightly larger deviation; we attribute this to vLCM's instruction-following training which introduces stylistic paraphrasing, rather than semantic drift.

**Ablation with the Round-trip Retrieval.** We further conduct an ablation study with the round-trip retrieval where we use three groups of captions (Groundtruth, V-SONAR Decoder, and V-LCM) as queries to retrieve the source videos in PE-Video. In Table 8, we find that Captions decoded by SONAR or LCM retrieve the correct video with extremely high accuracy. Notably, LCM is within 0.2% of SONAR on R@1. If cross-modal drift were substantial, retrieval accuracy would drop sharply; instead, it remains high, confirming its semantic preservation.

**Visualizing the Cross-modal Drift.** Finally, we plot the similarity between vision embeddings and the ground-truth caption embeddings versus the embeddings for V-SONAR and LCM captions in Figure 13 and Figure 12. SONAR-decoded captions show nearly identical (or slightly better) cosine similarity/distance compared to ground truth, indicating negligible cross-modal drift. vLCM captions show a slightly larger deviation; we attribute this to vLCM's instruction-following training which introduces stylistic paraphrasing, rather than semantic drift (verified in the next experiment). The points cluster also is generally around the $y = x$ line, directly showing no significant systematic semantic shift.

# I QUALITATIVE CASE

## I.1 IMAGE QUALITATIVE CASES FOR V-LCM

We present the qualitative cases for image captioning in Figure 14 and image question answering in Figure 15.

## I.2 VIDEO QUALITATIVE CASES FOR V-LCM

We present the qualitative cases for video captioning and question answering task in Figure 16.

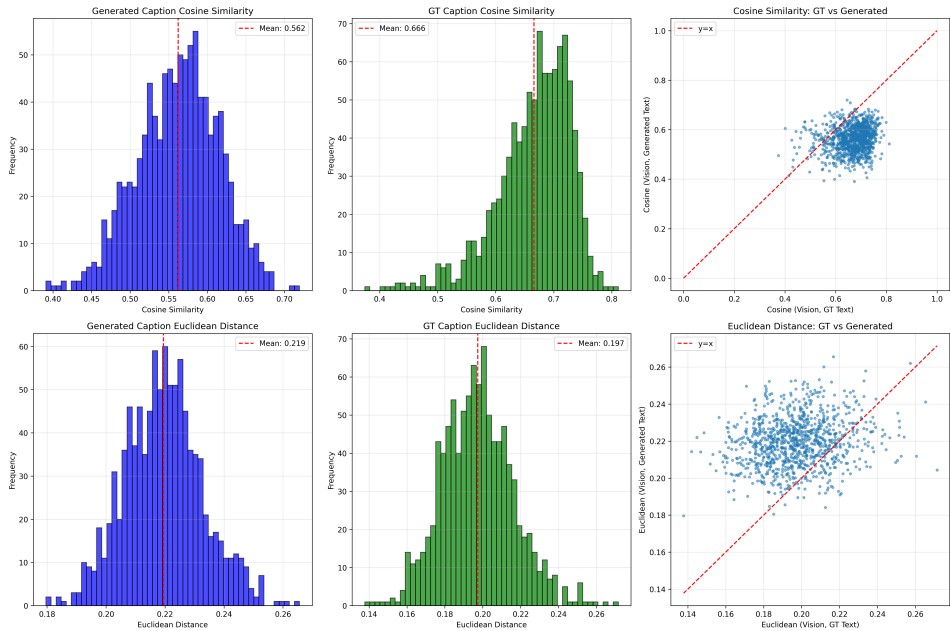

Figure 13: vLCM's visualization for cross-modal semantic drift.

| Vision Input | Prompt | Outputs |
|---|---|---|
| 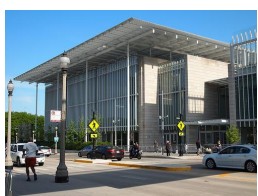 | Write a short and informative description that highlights the main subjects and actions occurring in the given image. `<IMAGE>` | **LCM**: *"The video shows a man riding a red-colored bike on a red-colored motorbike riding on a black mountain, with a motorcycle and a rug, and a red man standing behind a mountain with a bridge and a grass and a roof."* 
 **v-LCM**: *"A man riding a red motorcycle on a mountain."* |
| | Compose a descriptive paragraph that brings the image to life through detailed storytelling, connecting the various visual elements into a unified narrative. `<IMAGE>` | **LCM**: *"The vision begins with the bright video building standing in front of a broadly paved building, and a crowd of people walking on the street. As the sun shines over the building, the buildings begin to streak down the street. The vehicles move closer to the street, with the buildings creeping down the street. The vehicles are moving to the street, and there are people standing on the other side of the pavement. Eventually, the vision ends with the vehicles moving off onto the street, and the buildings slide down the street."* 
 **v-LCM**: *"A large building is walking near the corner of a street. There are white windows in front of the building. There is a large grey pole in front of the building. There are people walking on the street in front of the pole."* |

Figure 14: Qualitative examples for image captioning task in M3IT benchmark. Each row shows the input image, the given prompt, and the generated model output.

## I.3 QUALITATIVE CASES FOR VIDEO CAPTIONING BY v-SONAR

In addition to the qualitative cases for LCM. We present the qualitative cases for video captioning with v-SONAR compared to other VLMs in PVD-Bench and DREAM-1K from Figure 17 to Figure 20.

| Vision Input | Prompt | Outputs |
|---|---|---|
|  | Observe the image closely, recognize the named entities present, and use your knowledge to answer the question in relation to those entities. <IMAGE> Who plays at this stadium ? | **LCM**: *"The entities that play at this stadium are Red Stadium."*
**v-LCM**: *"The team that plays at this stadium is Manchester United."* |
|  | Analyze the image to locate any named entities and provide an appropriate response to the question. Start of vision. <IMAGE> End of vision. In this game, how many points is a 'V' tile worth? | **LCM**: *"It is not specified in the information provided that how many points a 'V' tile is worth."*
**v-LCM**: *"A 'V' tile is worth 4 points."* |
|  | Carefully review the question and the related image/text context, identify the precise answer from the multiple-choice selections, and provide a detailed, logical explanation to support your choice, illustrating the multi-jumbo reasoning journey.
<IMAGE>
Question: What of the following could Gordon's test show?
[...]
Options:
(A) if the spacecraft was damaged when using a parachute with a 1 m outlet going 200 km per hour.
(B) How stable was a parachute with an 1 m vent at 200 km per hour.
(C) whether a parachute with a 1 m vent would sway too much at 400 km/h. | **LCM**: *"(B) how stable was a parachute with a 1 m vent at 200 km per hour."*
**v-LCM**: *"The answer is (B) how stable was a parachute with a 1 m air leak was at 200 km per hour."* |
|  | Please respond to the query by referring to the picture and its limited entities. <IMAGE> What is the name of the translator? | **LCM**: *"The name of the translator is not provided."*
**v-LCM**: *"The name of the translator is Rebecca Herting."* |

Figure 15: Qualitative examples for image question answering tasks. Each row shows the input image, the given prompt, and the generated model output.

## I.4 QUALITATIVE CASES FOR VIDEO CAPTIONING BY V-SONAR

We present the qualitative cases for visual-commonsense-reasoning (VCR) with v-LCM compared to other VLMs in Figure 21. The qualitative example demonstrates that the aligned encoders, thought being trained with semantic-level caption, can still capture layout grounding and the spatial reasoning ability.

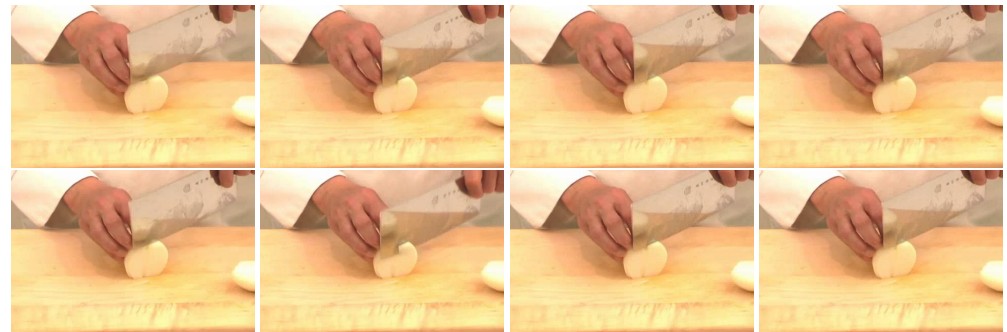

**Prompt:** You will be shown a video and asked a question about what you saw. Your answer should be a brief sentence that fits the video's contents. `<VIDEO>` what is a man slicing?
**LCM:** *A man is cutting an onion with a knife.*
**v-LCM:** *The man in the video is slicing an onion.*

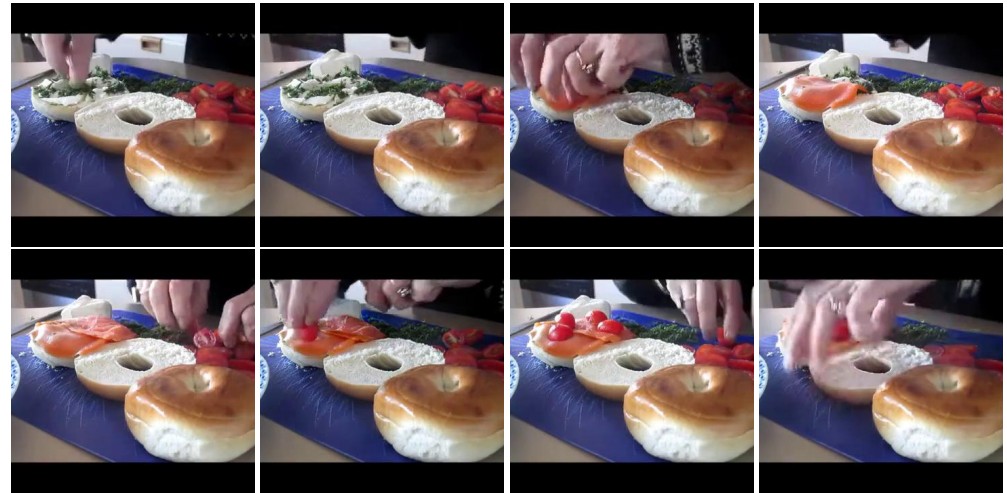

**Prompt:** Use your observation skills to answer the given question with the help of a short video clip. Start of vision. <VIDEO> End of vision. What type of bread is shown?
**LCM:** *The type of bread shown is a sandwich slice.*
**v-LCM:** *The bread shown in the video is a bagel.*

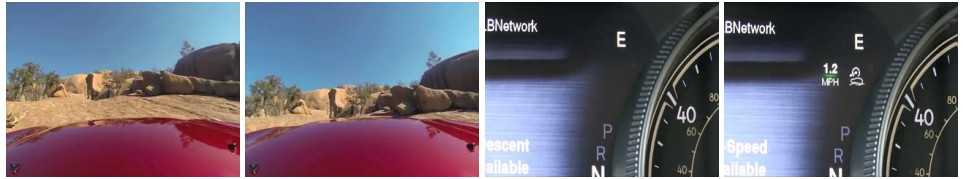

**Prompt:** Watch the video and create a caption based on the visual cues. `<VIDEO>`
**LCM:** *[End of Response.]*
**v-LCM:** *someone is driving a red car*

Figure 16: Qualitative examples for video captioning and question answering tasks. The frames are presented from left to right, and top to the bottom.

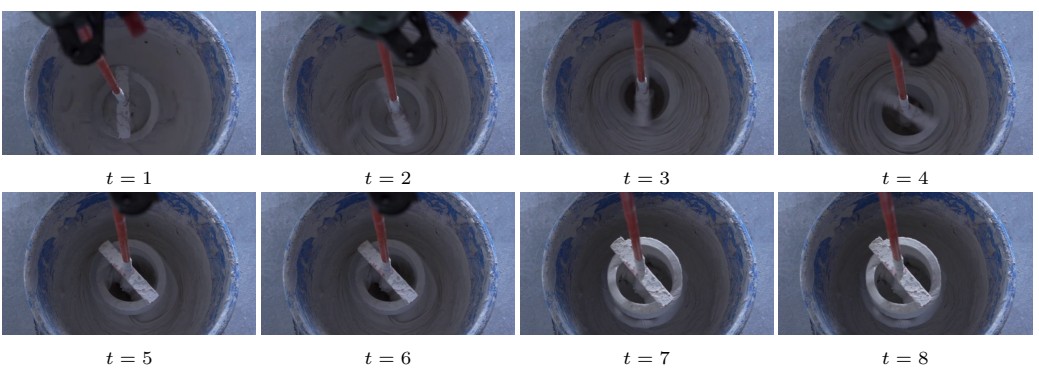

|  |  |  |  |
|---|---|---|---|
| $t = 1$ | $t = 2$ | $t = 3$ | $t = 4$ |
| $t = 5$ | $t = 6$ | $t = 7$ | $t = 8$ |

| Models | Video Captions |
|---|---|
| **V-SONAR (Ours)** | The video shows a **blue mixer** stirring a **white powder** in a round bucket. The bucket is moving with a rotating motion and is attached to a blue cylinder. The mixer is moving slowly in the middle, creating a white powder. |
| PLM-3B | In the background, there is a bucket, a stick, and a grey surface. The sound of the mixer is audible. *[Vague; mentions audio]* |
| Qwen2.5-VL-3B-Instruct | A close-up shot of a blue bucket filled with white paint. A **red-handled paintbrush** is dipped into the paint and then lifted out, leaving a trail of paint behind it. The brush is then lowered back into the paint and the process is **repeated several times**. The camera remains stationary. |
| InternVL2.5-1B | A **yellow hand** reaches into the white plastic bucket, grabbing the red cylindrical object, which is a tool for **pouring cement**. |

Figure 17: **Qualitative comparison on fine-grained motion understanding in PE-Video.** While state-of-the-art VLMs (Qwen-2.5, InternVL-2.5) suffer from hallucinations or captioning erors (highlighted in **red**) such as a "paintbrush" or "yellow hand," and PLM remains vague, **SONAR** accurately captures the mechanics of the mixer and the blue cylinder attachment (highlighted in **green**).

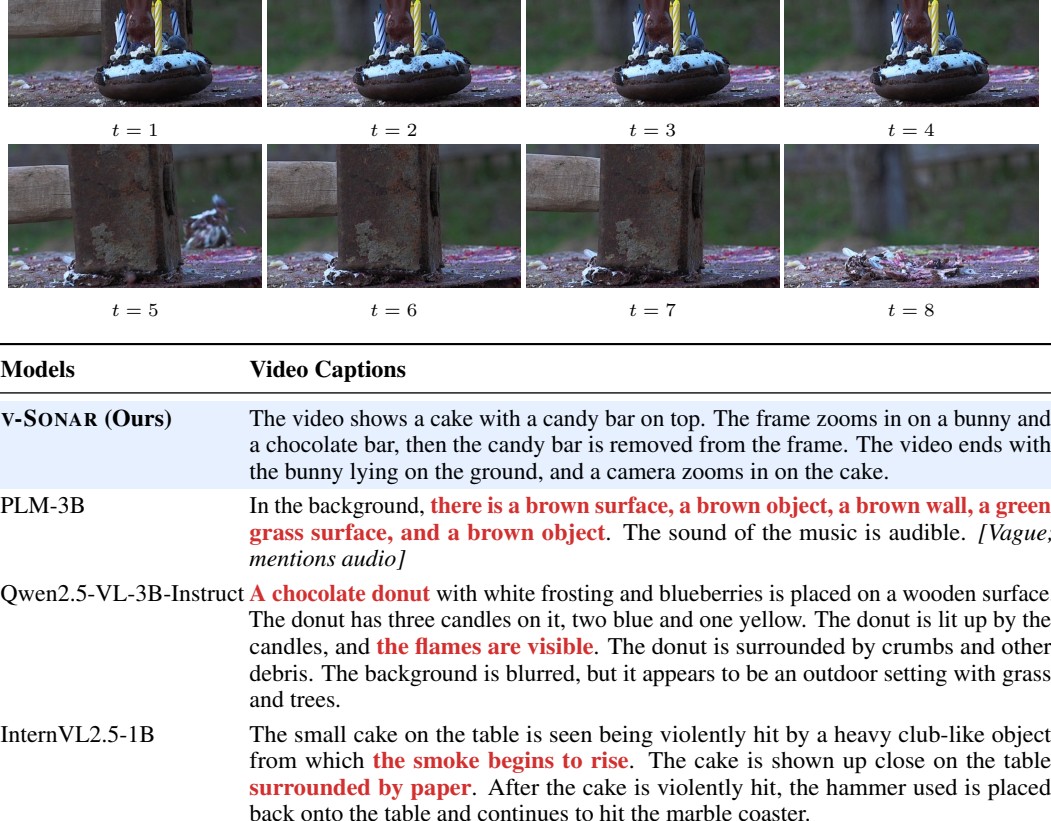

| Models | Video Captions |
|---|---|
| **V-SONAR (Ours)** | The video shows a cake with a candy bar on top. The frame zooms in on a bunny and a chocolate bar, then the candy bar is removed from the frame. The video ends with the bunny lying on the ground, and a camera zooms in on the cake. |
| PLM-3B | In the background, **there is a brown surface, a brown object, a brown wall, a green grass surface, and a brown object**. The sound of the music is audible. *[Vague; mentions audio]* |
| Qwen2.5-VL-3B-Instruct | **A chocolate donut** with white frosting and blueberries is placed on a wooden surface. The donut has three candles on it, two blue and one yellow. The donut is lit up by the candles, and **the flames are visible**. The donut is surrounded by crumbs and other debris. The background is blurred, but it appears to be an outdoor setting with grass and trees. |
| InternVL2.5-1B | The small cake on the table is seen being violently hit by a heavy club-like object from which **the smoke begins to rise**. The cake is shown up close on the table **surrounded by paper**. After the cake is violently hit, the hammer used is placed back onto the table and continues to hit the marble coaster. |

Figure 18: **Qualitative comparison in PE-Video.** We highlight the errors in **red**.

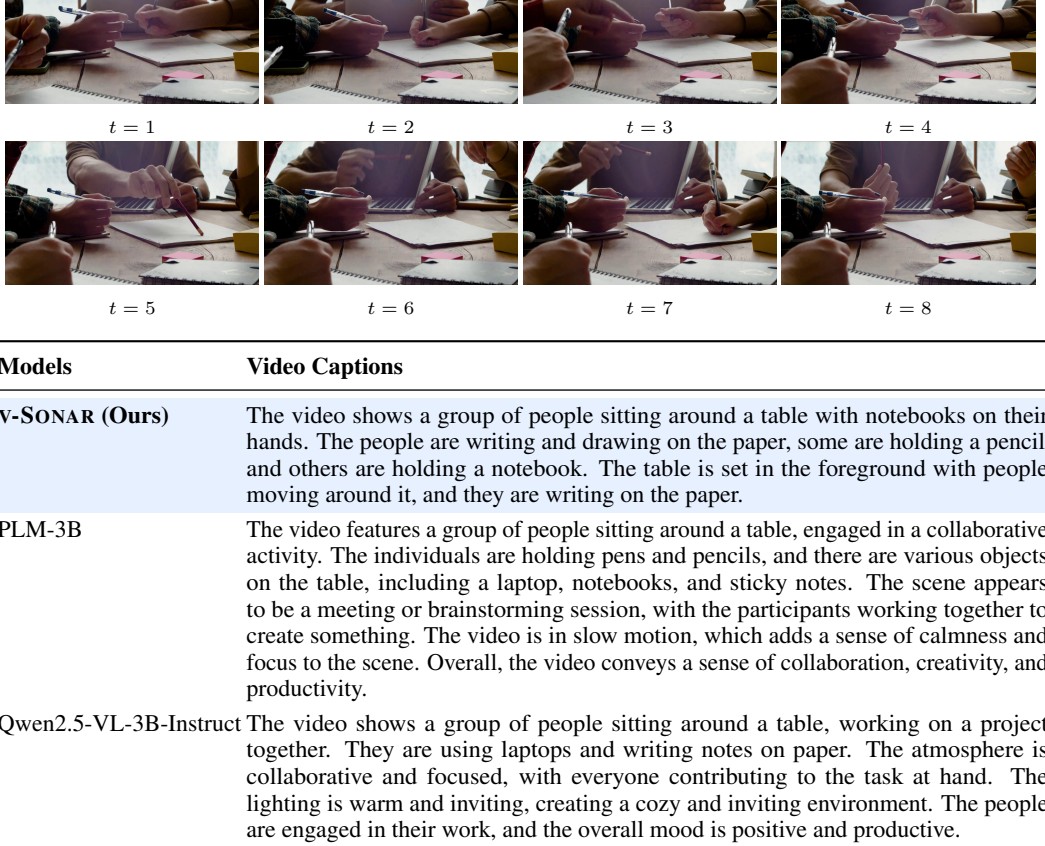

| Models | Video Captions |
|---|---|
| **V-SONAR (Ours)** | The video shows a group of people sitting around a table with notebooks on their hands. The people are writing and drawing on the paper, some are holding a pencil, and others are holding a notebook. The table is set in the foreground with people moving around it, and they are writing on the paper. |
| PLM-3B | The video features a group of people sitting around a table, engaged in a collaborative activity. The individuals are holding pens and pencils, and there are various objects on the table, including a laptop, notebooks, and sticky notes. The scene appears to be a meeting or brainstorming session, with the participants working together to create something. The video is in slow motion, which adds a sense of calmness and focus to the scene. Overall, the video conveys a sense of collaboration, creativity, and productivity. |
| Qwen2.5-VL-3B-Instruct | The video shows a group of people sitting around a table, working on a project together. They are using laptops and writing notes on paper. The atmosphere is collaborative and focused, with everyone contributing to the task at hand. The lighting is warm and inviting, creating a cozy and inviting environment. The people are engaged in their work, and the overall mood is positive and productive. |
| InternVL2.5-1B | In this video, a team of individuals collaborates to contribute to the design process on a desk. |

Figure 19: **Qualitative comparison in DREAM-1K.**

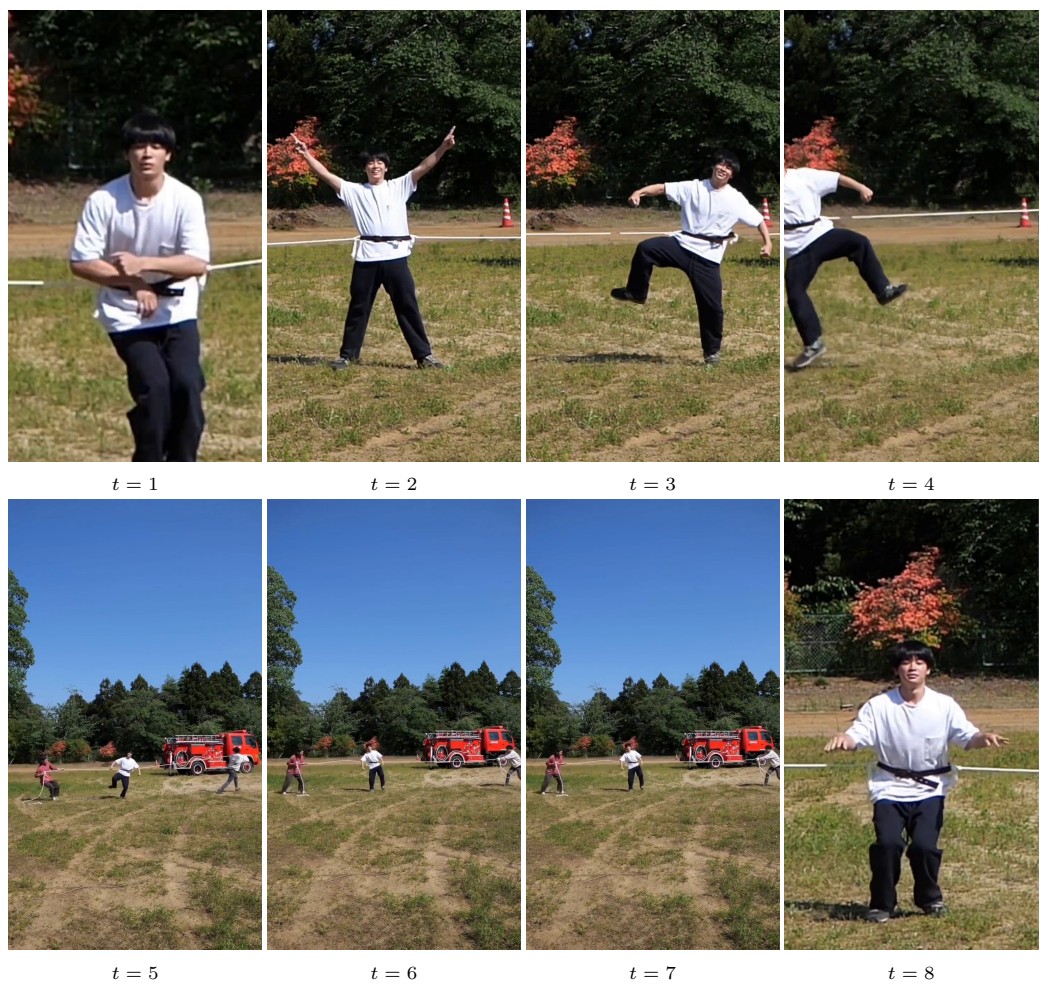

| Models | Video Captions |
|---|---|
| **V-SONAR (Ours)** | The video shows a group of men dancing in a white shirt. A man is standing in the field with a rope in his hands and a red truck behind him. The man is pulling the rope and the truck is moving towards the field. |
| PLM-3B | A man wearing a white t-shirt and black pants is standing on the grass surface and then starts running and kicking the white rope. Another man wearing a red t-shirt and black pants is standing on the grass surface and then starts running and kicking the white rope. The third man wearing a white t-shirt and black pants is standing on the grass surface and then starts running and kicking the white rope. In the background, there is a grass surface, trees, a red vehicle, a white rope, and the sky. The voice of the people is audible. |
| Qwen2.5-VL-3B-Instruct | The video shows a man standing in a field, wearing a white t-shirt and black pants. He is holding a rope and appears to be playing a game of tug of war. The background features trees and a clear blue sky. |
| InternVL2.5-1B | A performer in a white shirt and black belt balances on a long rope in a park, with a red vehicle and a lush red tree in the background. The performer changes their stance, eventually falling off the rope and landing on the ground, then quickly gets back up ready for the next try or battle. |

Figure 20: **Qualitative comparison in DREAM-1K.**

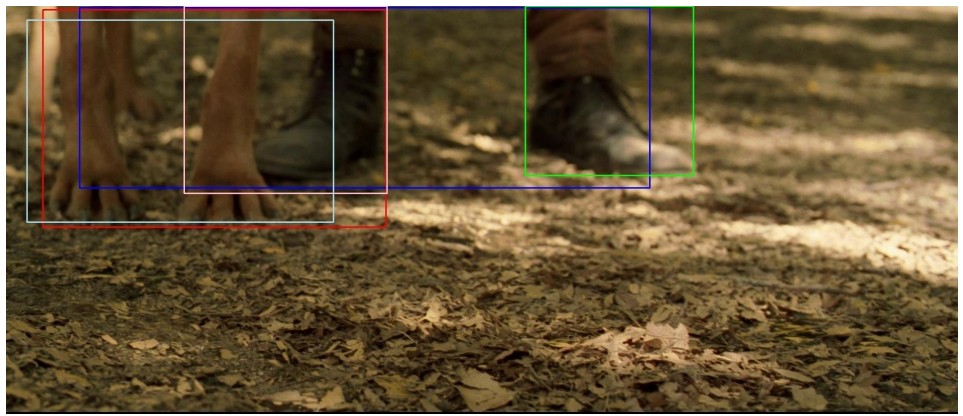

**Models**

| Prompt | Given an image and some bounded objects, answer the question and explains why the answer is right. Objects: The 0-th object is person in the red color box. The 1-th object is person in the blue color box. The 2-th object is person in the green color box. The 3-th object is person in the pink color box. The 4-th object is dog in the light blue color box. Question: What is the 3-th object doing with the 4-th object ? |
|---|---|
| PLM-8B | Young children enjoy taking their pets for a walk outside. |
| Qwen2.5-VL-8B-Instruct | Answer (A) the 3-th object is taking the 4-th object for a walk outside. Rationale: The 3-th object appears to be a child based on the size and posture, and the 4-th object is a dog. The context of the image suggests an outdoor setting with a child and a dog, which aligns with the idea of a child taking a pet for a walk. Dogs often enjoy walks, and it's common for young children to take their pets for walks as part of their daily routine or playtime. The other options do not fit the context of the image as well; there is no indication of flirting, looking at a computer, or setting up anything. |
| InternVL2.5-8B | the 3-th object was setting up the 4-th object. |
| v-LCM | The 3rd object is taking the 4th object for a walk outside. 'Cause little kids like to take their pets for a walk. |

Figure 21: **Qualitative comparison in Visual-Commonsense-Reasoning (VCR),** which requires grounding and spatial reasoning ability connecting with the commonsense.

