# OpenReview forum: "Unified Vision–Language Modeling via Concept Space Alignment"
_ICLR.cc/2026/Conference — ICLR 2026 Poster_

### Official Review · Reviewer_wLL2 · 2025-10-25

**Soundness:** 3
**Presentation:** 4
**Contribution:** 3
**Rating:** 6
**Confidence:** 3

**Summary:**

This paper introduces v-Sonar, a paradigm to map vision representation into Sonar’s embedding space via representation alignment, enabling direct reasoning and decoding in a shared latent space. v-Sonar achieved strong zero-shot text-to-video retrieval and video captioning performance. Building upon this, the paper introduces v-lcm, a multi-modal version of large concept models, which perform latent diffusion over unified visual-text latent space, and achieves comparable performance to SOTA VLMs.

**Strengths:**

1. The paper is well-written and easy to follow.
2. The evaluation and ablation studies are very comprehensive.
3. v-Sonar and v-lcm present great zero-shot capabilities, which are very interesting.

**Weaknesses:**

1. The paper is more like a technique report with limited technical contributions. The core concept, aligning vision encoders to text encoders, has been proposed by previous work [1]. The specific techniques to align visual-textual representations (MSE and contrastive loss) are also very standard.
2. Most metrics are only comparable to SOTA models, except for the multilingual ability, which may partly be due to the fact that Sonar is specially designed and trained with specific datasets for this objective. While it is encouraging to explore architectures different from current LLMs even with temporary underperformance, it is preferable that the new model can demonstrate significant advantages in certain special aspects.
3. Some of the implementation details are unclear. See questions.

[1] Jose, Cijo, et al. "Dinov2 meets text: A unified framework for image-and pixel-level vision-language alignment." CVPR 2025.

**Questions:**

1. The open-sourced PE-VIDEO dataset contains around 120k videos with annotation. However, v-Sonar uses 200k video-caption pairs to refine the alignment in stage 3 and uses 15k pairs for evaluation. Are there any overlaps?

2. How is v-lcm supervised finetuned? Do you update all the parameters?

---

> ### Author Response · Authors · 2025-11-21
> **Response to Reviewer wLL2**
>
> We would like to thank the reviewer for the thorough review. Please find below our answers to your questions.
>
> **1. Technical Contributions and Distinction from Prior Work**
>
> > **Review:** The paper is more like a technique report with limited technical contributions. The core concept, aligning vision encoders to text encoders, has been proposed by previous work (Dinov2 meets text) [1]. ...
>
> **Response:** We appreciate the opportunity to clarify the distinctions between our work and "Dino Meets Text" (and similar alignment frameworks). Our contribution differs in two fundamental ways:
>
> 1.  **Direction of Alignment (Vision $\to$ Text Encoder-Decoder):** "Dino Meets Text" aligns a new text encoder to a frozen vision encoder. Our approach reverses this: we train an off-the-shelf vision encoder (Perception Encoder) to align with a frozen language/modality-agnostic encoder–decoder model (SONAR). **This inversion is critical because it allows the SONAR decoder to decode visual embeddings directly into the 200 languages supported by SONAR, while enabling the introduction of V-LCM.** To make the alignment work, we perform the large-scale experiments, including carefully ablating the encoder architecture, data pipeline, learning objective as in Sec3.2 and Appendix B.
> 2.  **Evaluation through Captioning:** To the best of our knowledge, no prior post-hoc alignment method has evaluated vision–text alignment by **decoding** visual embeddings through a pretrained text decoder into captions. This provides a far more stringent test of semantic alignment compared to standard retrieval-only evaluations.
>
> **2. SOTA Comparison and Advantages of the New Paradigm**
>
> > **Review:** Most metrics are only comparable to SOTA models... it is preferable that the new model can demonstrate significant advantages in certain special aspects.
>
> **Response:** We respectfully clarify that our objective is not merely to surpass all SOTA vision–language models on English benchmarks, but to validate a fundamentally different modeling paradigm. Existing VLMs (e.g., Qwen-VL, InternVL) operate at the discrete token level. In contrast, V-LCM performs multimodal reasoning entirely in a language/modality-agnostic **continuous space** with a diffusion objective. This paradigm yields significant advantages in specific aspects:
>
> 1.  **Zero-shot Cross-Modality Generalization:** An LCM trained on **text-only** data can process visual inputs from V-SONAR without retraining. On PE-VIDEO, DREAM-1K, and VATEX, the zero-shot LCM performs within 1–5 BLEU of multimodally-trained 8B-scale baselines (Table 5), despite never seeing video data during training.
> 2.  **Multilingual Generalization (200 Languages):** Because both SONAR and V-SONAR are trained in a language-agnostic space, V-LCM demonstrates superior instruction-following ability in **61 out of 62 tested languages** in the M3IT multilingual evaluation (Fig. 3). It outperforms Qwen2.5-VL-7B and PLM-8B, particularly in low-resource languages such as Burmese and Tajik.
> 3.  **Unified Training Interface:** V-LCM maintains a single latent diffusion objective across modalities, avoiding the architectural divergence common in VLMs. This design is conceptually closer to world-modeling in concept space than to standard VLM pretraining.
>
> While our results are competitive rather than dominant in English benchmarks, they demonstrate robust generalization and a scalable path to multilingual multimodality.
>
> **3. Clarification on Data Overlap**
>
> > **Review:** The open-sourced PE-VIDEO dataset contains around 120k videos... v-Sonar uses 200k video-caption pairs... and uses 15k pairs for evaluation. Are there any overlaps?
>
> **Response:** We confirm that there is **no overlap** between the training and evaluation data.
> *   **Training:** The 200k high-quality video–caption pairs used in Stage 3 are part of the training set.
> *   **Evaluation:** Conducted exclusively on the PE-VIDEO Benchmark (PVD-Bench), which is a 15k human-refined, **held-out test set**.
>
> We will clarify this distinction explicitly in the revised manuscript.
>
> **4. Supervision and Parameter Updates**
>
> > **Review:** How is v-lcm supervised finetuned? Do you update all the parameters?
>
> **Response:** During V-LCM’s supervised fine-tuning on M3IT:
> *   **Frozen:** All encoders (SONAR and V-SONAR) and the SONAR decoder remain completely frozen.
> *   **Trainable:** Only the **LCM diffusion backbone** is updated under the next-embedding prediction objective.
>
> This ensures that multimodal instruction tuning affects only the generative backbone, preserving the pretrained multilingual and multimodal structure of SONAR and V-SONAR. We will add this clarification to Section 3.4.

---

> > ### Comment · Reviewer_wLL2 · 2025-11-26
> >
> > I thank the authors for their responses. I have raised my score.

---

> > > ### Author Response · Authors · 2025-11-28
> > >
> > > Dear reviewer wLL2,
> > >
> > > We appreciate your careful evaluation of our work and the rebuttal, and for raising the score; we value your insights and the time you invested in our submission.
> > >
> > > Best,
> > >
> > > Authors

---

### Official Review · Reviewer_fuEm · 2025-10-30

**Soundness:** 3
**Presentation:** 3
**Contribution:** 2
**Rating:** 6
**Confidence:** 2

**Summary:**

The paper extends the SONAR multilingual embedding space to vision via V-SONAR, which aligns a pretrained Perception Encoder to SONAR through a post-hoc MSE-based mapping and a three-stage image/video captioning curriculum. This enables the Large Concept Model (LCM), trained only on text embeddings, to operate directly on visual inputs. The authors further introduce V-LCM, a vision–language instruction-tuned version trained in the shared SONAR/V-SONAR latent space. The approach achieves zero-shot video retrieval and captioning, and V-LCM matches or surpasses state-of-the-art models on multilingual vision–language tasks across most of the languages.

**Strengths:**

1. The paper discussed a clear and well-motivated idea of extending the SONAR embedding to visual modalities through a lightweight post-hoc alignment.
2. The proposed alignment pipeline is designed well and experimentally validated.

**Weaknesses:**

1. The paper’s main idea, while clearly implemented, feels somewhat incremental as it builds on existing post-hoc alignment frameworks rather than introducing a fundamentally new alignment principle.

2. The evaluation mainly focuses on retrieval and captioning benchmarks, which measure semantic alignment but not grounding quality. It remains unclear whether V-SONAR preserves sufficient spatial and compositional information for tasks that require reasoning about object relations, layouts, or fine-grained attributes.

3. The reported numerical gains cannot be attributed solely to the proposed alignment, since the results combine the benefits of the stronger SONAR2 text space and the powerful Perception Encoder backbone. A more controlled comparison, such as aligning to both SONAR1 and SONAR2, or directly comparing against the unaligned vision encoder, would better isolate the contribution of the alignment itself.

**Questions:**

- Can the authors report a controlled comparison, such as aligning to SONAR1 versus SONAR2 or evaluating against the unaligned Perception Encoder, to quantify the gain contributed by the proposed alignment?

---

> ### Author Response · Authors · 2025-11-21
> **Rebuttal response to reviewer fuEm**
>
> We would like to thank the reviewer for the thorough review. Please find below our answers to your questions.
>
> **1. Novelty and "Incremental" Nature of the Approach**
>
> > **Review:** The paper’s main idea... feels somewhat incremental as it builds on existing post-hoc alignment frameworks rather than introducing a fundamentally new alignment principle.
>
> **Response:** We explore an alternative vision-language modeling approach which operates in a language/modality-agnostic space curated by V-SONAR and SONAR. We argue that our approach has fundamental differences compared to prior work:
>
> 1. **Wide coverage in modalities and languages:** V-SONAR is the first extension of a universal text and speech embedding space (SONAR, supporting 200 text/37 speech languages) to image and video modalities. This unification makes SONAR the most universal embedding space to date, spanning four modalities.
>
> 2. **Post-hoc alignment to a decoder space:** To the best of our knowledge, we are the first to conduct a large-scale effort to align a vision encoder with a textual *encoder-decoder* embedding space via post-hoc alignment.
>
> 3. **New paradigm in latent space modeling:** Our work introduces V-LCM, which unifies vision and language into a single sequence of latent embeddings. This represents a new paradigm: fusing modalities in a modality-agnostic latent space *prior* to input, enabling autoregressive generation via a latent diffusion objective.
>
> 4. **Zero-Shot Multimodality Transfer:** This alignment strategy allows the text-only trained LCM to perform zero-shot single- and multi-visual concept understanding tasks, such as video captioning and summarization. This conceptual transfer is a significant finding.
>
> **2. Evaluation of Grounding and Spatial Reasoning**
>
> > **Review:** It remains unclear whether V-SONAR preserves sufficient spatial and compositional information for tasks that require reasoning about object relations, layouts, or fine-grained attributes.
>
> **Response:** We wish to highlight that our evaluation extends well beyond simple captioning and retrieval, explicitly measuring concept understanding and reasoning via the V-LCM model:
>
> *   **Multi-Concept Reasoning:** Section 3.3 includes Long Video Summarization on VIDEOXUM, a task requiring the model to ground over a sequence of multiple visual embeddings (multiple V-SONAR embeddings).
> *   **Visual Question Answering (QA):** In Section 3.4, V-LCM is evaluated on multiple QA tasks in the M3IT benchmark, including VisualMRC, VIQUAE, IVQA, MSRVTT-QA, and ActivityNetQA. These generation and multiple-choice tasks require understanding of object relations and layout-based reasoning (VisualMRC), temporal grounding (MSRVTT-QA, ActivityNetQA), and commonsense reasoning (IVQA, ScienceQA, VIQUAE).
> *   **Grounding & Spatial Reasoning Benchmark:** We also provide the vLCM results against other models of comparable size on the visual-commonsense-reasoning benchmark. This benchmark explicitly evaluates layout grounding and spatial reasoning (see Figure 17 in Appendix I). Models must reason about objects using commonsense knowledge while interpreting their bounding boxes. We report the token-level F1 score compared to the groundtruth, and semantic similarity compared to the reference rationale. The strong vLCM performance demonstrates that, despite being trained only with semantic-level captions, it still effectively learns layout grounding and spatial reasoning.
>
> | VCR           | F1    | Sim. |
> |---------------|-------|---------------------|
> | LCM           | 0.385 |               0.258 |
> | vLCM          | 0.671 |               0.529 |
> | PLM-8B        | 0.441 |               0.432 |
> | Qwen-2.5-7B   | 0.275 |               0.402 |
> | Qwen-2-7B     | 0.502 |               0.513 |
> | InterVL2.5-8B | 0.155 |               0.158 |
> | InterVL2-8B   | 0.325 |               0.340 |
>
>
> *   **Fine-grained Visual Feature Preservation:** The ablation in Figure 2 demonstrates that V-SONAR embeddings retain richer visual features for reasoning than text-only representations (re-encoded captions by SONAR), as LCM with V-SONAR consistently outperforms LCM with only SONAR inputs across short, mid, and long videos.
>
> These results empirically validate that V-SONAR preserves sufficient semantic and relational information for complex vision-language reasoning.

---

> > ### Author Response · Authors · 2025-11-21
> >
> > **3. Controlled Comparison (Alignment vs. Backbone)**
> >
> > > **Review:** Can the authors report a controlled comparison, such as aligning to SONAR1 versus SONAR2 or evaluating against the unaligned Perception Encoder, to quantify the gain contributed by the proposed alignment?
> >
> > **Response:** We would like to point out that the paper already provides three explicit points of comparison that isolate the contribution of the alignment:
> >
> > **A. Comparison to Unaligned Vision Encoder (PE).**
> > PECoreG in Table 2 represents the unaligned Perception Encoder. We demonstrate that V-SONAR alignment preserves strong zero-shot retrieval capability with only a minor performance drop on PE-VIDEO (0.04 drop in Recall@1) and a gain on VATEX (1.3 point gain in Recall@1). Since the goal of alignment is not to improve the vision encoder itself, but rather to equip it with SONAR advantages (e.g., multilinguality, decoding capability), this minor drop is acceptable.
> >
> > **B. Comparison of SONAR1 vs. SONAR2.**
> > We have aligned the Perception Encoder to both SONAR1 and SONAR2 (detailed in Appendix E and Figure 4):
> > *   **We use SONAR 1 and SONAR2 for verifying different hypothesis:** SONAR2 was used only for proving the feasibility of aligning an existing vision encoder with a textual encoder-decoder model. We verify this by SONAR2 encoder in retrieval task, and SONAR2 decoder in captioning task. We then use SONAR1 to verify for leveraging it with LCM because the original **LCM backbone was trained exclusively on SONAR1 embeddings**.
> > *   **Captioning Results:** As shown in Figure 4, **even with SONAR1, performance is competitive with existing VLMs** (PVD-Bench: 35 R-L vs. second-best 31.2; DREAM-1K: 21 R-L vs. second-best 20.4). Changing from SONAR2 to SONAR1 does not qualitatively change our conclusions.
> >
> > **C. Additional Retrieval Comparison (SONAR1 vs. SONAR2).**
> > To further address the reviewer's request, we provide the retrieval comparison between SONAR1 and SONAR2 in addition to the provided captioning results. While SONAR2 generally holds an advantage, SONAR1 remains highly competitive (e.g., R@1 of 64.9 on PVD-Bench compared to PECoreG).
> >
> > | Dataset | Model | R@1 | R@5 | R@10 | MRR |
> > | :--- | :--- | :--- | :--- | :--- | :--- |
> > | **PVD-Bench** | SONAR1 | 0.649 | 0.843 | 0.895 | 0.737 |
> > | | SONAR2 | 0.666 | 0.860 | 0.909 | 0.753 |
> > | **DREAM-1K** | SONAR1 | 0.536 | 0.759 | 0.838 | 0.638 |
> > | | SONAR2 | 0.469 | 0.694 | 0.767 | 0.570 |
> > | **VATEX** | SONAR1 | 0.119 | 0.263 | 0.350 | 0.195 |
> > | | SONAR2 | 0.202 | 0.414 | 0.506 | 0.304 |
> >
> > Finally, **Table 4** in the Ablation Study quantifies the exact contribution of our curriculum alignment strategy (SV, IC) and projector architecture to the final performance.

---

### Official Review · Reviewer_XhVS · 2025-10-30

**Soundness:** 3
**Presentation:** 3
**Contribution:** 3
**Rating:** 6
**Confidence:** 4

**Summary:**

The paper introduces V-SONAR, a framework that aligns visual encoders into the SONAR multilingual concept space, and extends the Large Concept Model (LCM) to vision–language tasks via V-LCM.
Instead of joint multimodal training, the authors propose a post-hoc alignment of frozen visual features to an existing multilingual embedding space, enabling zero-shot and multilingual image/video understanding.
Extensive experiments on retrieval, captioning, and multilingual instruction following show competitive or superior results, especially across 61 of 62 evaluated languages.

**Strengths:**

- **Elegant unification:** The method merges vision and language in a shared continuous embedding space, avoiding heavy multimodal retraining.

- **Practical efficiency:** Simple MSE-based alignment achieves strong zero-shot and multilingual performance with modest compute.

- **Comprehensive evaluation:** Covers image/video tasks, long-video summarization, and multilingual benchmarks.

- **Insightful analysis:** Provides embedding-geometry diagnostics (AC/trace/logdet) to interpret cross-modal alignment quality.

- **Strong empirical results:** Outperforms major baselines like SigLIP2 and PLM on several metrics.

**Weaknesses:**

- **Lack of clear conceptual  motivation** –
The paper reads more like a comprehensive technical report. While the system is well engineered, the underlying motivation and novelty are not sharply defined. The central idea—post-hoc alignment of visual features to a frozen multilingual embedding space—feels incremental, relying on existing components (SONAR, LCM).

- **Potential semantic collapse within the aligned subspace.**
Aligning visual features via an MSE loss to a frozen text space risks over-compression: different visual scenes may map to similar embeddings. The paper lacks an investigation into embedding diversity or anisotropy after alignment—important for understanding retrieval and generation limits.

- **No examination of cross-modal consistency at inference.**
The paper shows that visual inputs can be decoded by the LCM text generator, but doesn’t analyze whether the generated language faithfully represents the same semantic region of the embedding space. Cross-modal drift—where decoding shifts meaning—may exist but is untested.

**Questions:**

- When decoding visual inputs into text via LCM, have you evaluated whether the generated descriptions remain semantically faithful to the corresponding visual embeddings? If not, how do you assess or mitigate possible cross-modal drift?
- Could you include some qualitative or more interpretable examples (e.g., generated captions, retrieval results, or typical errors) to illustrate how the model behaves in practice?
- Could the post-hoc alignment to a fixed text space cause semantic drift or information loss, especially for fine-grained visual details that have no direct linguistic counterpart?

---

> ### Author Response · Authors · 2025-11-21
> **Response to Reviewer XhVS**
>
> We would like to thank the reviewer for the thorough review. Please find below our answers to your questions.
>
> **1. Conceptual Motivation and Novelty**
>
> > **Review:** Lack of clear conceptual motivation – The paper reads more like a comprehensive technical report.
>
> **Response:** We respectfully argue that our work explores a fundamentally alternative vision-language modeling approach which operates in a language/modality-agnostic space curated by V-SONAR and SONAR. This is distinct from prior work in four key aspects:
>
> 1.  **Wide coverage in modalities and languages:** V-SONAR is the first extension of a universal text and speech embedding space (SONAR, supporting 200 text/37 speech languages) to image and video modalities. This unification makes SONAR the most universal embedding space to date, spanning four modalities.
> 2.  **Post-hoc alignment to a decoder space:** To the best of our knowledge, we are the first to conduct a large-scale effort to align a vision encoder with a textual **encoder-decoder** embedding space via post-hoc alignment. And we conduct the evaluation of vision-language alignment with captioning tasks rather than the retrieval-only setup.
> 3.  **New paradigm in latent space modeling:** Our work introduces V-LCM, which unifies vision and language into a single sequence of latent embeddings. This represents a new paradigm: fusing modalities in a modality-agnostic latent space *prior* to input, enabling autoregressive generation via a latent diffusion objective.
> 4.  **Zero-Shot Multimodality Transfer:** This alignment strategy allows the text-only trained LCM to perform zero-shot single- and multi-visual concept understanding tasks, such as video captioning and summarization. This conceptual transfer is a significant finding.
>
> **2. Embedding Diversity (Addressing Semantic Collapse)**
>
> > **Review:** Potential semantic collapse during alignment... The paper lacks an investigation into embedding diversity or anisotropy after alignment.
>
> **Response:** The reviewer correctly identifies the critical need to maintain embedding diversity post-alignment. We confirm that this investigation was conducted and reported in **Table 2**.
>
> We explicitly measure the spread of representations using the **Trace** of the covariance matrix and the **approximated volume** using log-determinant.
>
> *   **Quantitative Evidence:** As shown in Table 2, V-SONAR's vision embeddings (Vision Trace: **0.49 / 0.36 / 0.35**) maintain a highly competitive spread compared to state-of-the-art vision encoder baselines (e.g., SigLIP2 Vision Trace: 0.39 / 0.40 / 0.35).
> *   **Conclusion:** This indicates **no significant semantic collapse** compared to existing vision encoders. Furthermore, by freezing the highly expressive SONAR space, V-SONAR achieves the largest textual embedding dispersion (Text Trace and Logdet) among all compared models, suggesting successful alignment to a rich target space.

---

> ### Author Response · Authors · 2025-11-21
>
> **3. Cross-Modal Consistency at Inference**
>
> > **Review:** When decoding visual inputs into text via LCM, have you evaluated whether the generated descriptions remain semantically faithful to the corresponding visual embeddings?
>
> **Response:** Thank you for raising this important point. While our strong retrieval and captioning performance implies semantic alignment, we performed three dedicated analyses on 1,000 randomly sampled examples from PVD-Bench to directly measure semantic fidelity.
>
> **A. Embedding-Level Semantic Fidelity**
> For each video embedding $v$, we compare its similarity to: (i) ground-truth caption embedding ($t_{gt}$), (ii) SONAR Decoder caption embedding ($t_{sonar}$), and (iii) LCM caption embedding ($t_{lcm}$).
>
> | Comparison Target | Cosine Similarity to $v$ | Euclidean Dist. to $v$ |
> | :--- | :--- | :--- |
> | **Groundtruth Captions** | 0.666 | 0.197 |
> | **SONAR Decoder** | 0.688 | 0.175 |
> | **vLCM** | 0.562 | 0.219 |
>
> *   **Findings:** SONAR-decoded captions show nearly identical (or slightly better) cosine similarity/distance compared to ground truth, indicating **negligible cross-modal drift**. vLCM captions show a slightly larger deviation; we attribute this to vLCM’s instruction-following training which introduces stylistic paraphrasing, rather than semantic drift (verified in the next experiment).
>
> **B. Retrieval Round-Trip: Using Generated Captions to Retrieve Videos**
> We used the three groups of captions as queries to retrieve the source videos in PE-Video.
>
> | Query Source | R@1 | R@5 | R@10 | MRR |
> | :--- | :--- | :--- | :--- | :--- |
> | **Groundtruth Captions** | 87.0% | 95.9% | 97.1% | 0.908 |
> | **SONAR Decoder** | 82.5% | 97.0% | 98.7% | 0.888 |
> | **LCM** | 82.3% | 96.7% | 97.9% | 0.887 |
>
> *   **Findings:** Captions decoded by SONAR or LCM retrieve the correct video with extremely high accuracy. Notably, LCM is within 0.2% of SONAR on R@1. If cross-modal drift were substantial, retrieval accuracy would drop sharply; instead, it remains high, confirming its semantic preservation as discussed in experiment A.
>
> **C. Drift Visualization**
> We plotted $cos(v, t_{gt})$ vs. $cos(v, t_{sonar})$ for each example (added to Appendix H). The points cluster is generally around the $y=x$ line, directly showing no significant systematic semantic shift.
>
> **4. Qualitative Examples**
>
> > **Review:** Could you include some qualitative or more interpretable examples... to illustrate how the model behaves in practice?
>
> **Response:** We have included comprehensive qualitative cases in **Appendix I**, covering:
> *   Image captioning and paragraph generation
> *   Image-based QA and Commonsense reasoning
> *   OCR-based QA
> *   Video QA from V-LCM
>
> However, we do agree to additionally incorporate the following qualitative case:
> *   Video captioning by SONAR decoder compared to other VLMs in Figure 17-20.
> *   Grounding and spatial reasoning by vLCM compared to other VLMs in Figure 21.
>
> **5. Information Loss via Alignment**
>
> > **Review:** Could the post-hoc alignment to a fixed text space cause semantic drift or information loss, especially for fine-grained visual details?
>
> **Response:** We provide direct evidence in **Figure 2** that V-SONAR embeddings retain *more* visual information than the textual SONAR embeddings they are aligned to.
>
> On the long video summarization task (VIDEOXUM), LCM performs significantly better when using **raw V-SONAR embeddings** than when using SONAR embeddings generated from textual inputs (decoding and re-encoding). Moreover, this performance gap increases with video length. This confirms that the aligned vision encoder (V-SONAR) preserves rich, modality-specific visual features that purely linguistic captions cannot capture.
>
> **We add an additional result for vLCM results on VCR where models must reason about objects using commonsense knowledge while interpreting their bounding boxes. VCR requires significantly more visual information such as grounding and layout than semantic-level alignment (see Figure 17 in Appendix I).** We report the token-level F1 score compared to the groundtruth, and semantic similarity compared to the reference rationale. The strong vLCM performance demonstrates that, despite being trained only with semantic-level captions, it still effectively learns layout grounding and spatial reasoning.
>
> | VCR            | F1  |  Sim.  |   |
> |------------------|-------|-------|-------|
> | LCM              | 0.385 | 0.258 |       |
> | vLCM             | 0.671 | 0.529 |       |
> | PLM-8B           | 0.441 | 0.432 |       |
> | Qwen-2.5-7B      | 0.275 | 0.402 |       |
> | Qwen-2-7B        | 0.502 | 0.513 |       |
> | InterVL2.5-8B    | 0.155 | 0.158 |       |
> | InterVL2-8B      | 0.325 | 0.340 |       |

---

### Official Review · Reviewer_wU7h · 2025-11-01

**Soundness:** 3
**Presentation:** 3
**Contribution:** 2
**Rating:** 4
**Confidence:** 4

**Summary:**

This paper introduces v-SONAR, a vision-language embedding space constructed by post-hoc aligning a state-of-the-art vision encoder with the existing multilingual text embedding space Sonar. The authors propose a three-stage curriculum that leverages large-scale image-caption and video-caption datasets to align Perception Encoder representations into Sonar's semantic concept space, supporting 200 text languages and 37 speech languages in addition to images and videos. Thorough experiments evaluate the alignment and demonstrate competitive or superior zero-shot performance in text-to-video retrieval, video captioning, and especially in multilingual and multi-modal instruction following. The framework is further extended with v-LCM, a latent-diffusion-based unified vision-language model, which is instruction-tuned and evaluated across a large suite of language and vision-language downstream tasks.

**Strengths:**

1.Instead of training a massive multilingual VLM from scratch, the authors cleverly "inherit" the advantages of two SOTA models: SONAR's powerful multilingual semantic space and PERCEPTION ENCODER's strong visual representation capabilities. This approach is more computationally efficient and highly scalable.

2.By modeling in the unified SONAR/V-SONAR latent space, v-LCM successfully generalizes its visual understanding capabilities to a massive number of low-resource languages, significantly outperforming strong baselines like Qwen-VL and PLM-8B in 61 out of 62 languages.

3.The proposed Large Concept Model (LCM), trained only on text, is capable of directly processing V-SONAR encoded visual embeddings and performing complex tasks like video summarization without having seen any visual data. This strongly demonstrates that V-SONAR has successfully aligned visual concepts to the SONAR concept space at a semantic level.

4.The alignment method employs a three-stage coarse-to-fine curriculum, progressively achieving visual-semantic alignment. This training strategy demonstrates good engineering feasibility.

**Weaknesses:**

1.Critical Flaw: Inconsistent and Non-Reproducible SONAR Versions. The paper's two main SOTA (state-of-the-art) achievements are based on two different, incompatible SONAR versions. The SOTA video captioning (Table 3) uses SONAR2, an unreleased, higher-performing internal model . Conversely, the flagship multilingual SOTA results (Fig. 3) rely on the public SONAR1 , as the LCM was trained on it. This mismatch creates significant confusion about the paper's true contribution and renders the SOTA video captioning results non-reproducible.

2.The paper introduces v-LCM as a "new paradigm" built upon the Large Concept Model (LCM), which is a latent diffusion model . Unlike standard Transformers, diffusion models require multiple iterative sampling steps for generation, which is typically orders of magnitude slower. The paper provides no discussion on inference latency or computational cost, making it impossible to evaluate the practical utility of v-LCM against standard auto-regressive VLMs.

3.Lack of Analysis on Training Data Bias. The 12M+ image and video pairs used for alignment are likely English-centric. The paper provides no analysis of this potential data bias or how it impacts the model's outstanding multilingual generalization. It is unclear if this capability is learned during alignment or simply a "free lunch" inherited from the pre-trained SONAR space.

4.Multilingual Generalization vs. English SOTA. While the multilingual results are exceptional, the paper's own data (Table 5) shows that v-LCM is not the overall best on competitive English benchmarks. It trails the PLM-8B baseline in tasks like video summarization (VIDEOXUM) and document QA (VisualMRC). This indicates the model's core strength is its broad multilingual generalization rather than peak performance in high-resource English tasks.

5.The paper justifies its use of MSE loss over a standard contrastive loss by claiming "no significant gains" . However, the data in Appendix B (Table 6) shows a clear trade-off: contrastive loss performed better on retrieval (R@1 52.4 vs. 49.0) but slightly worse on captioning (BLEU 38.6 vs. 38.9). The paper lacks a necessary technical analysis of why this trade-off exists and why MSE is the better choice for generative alignment.

**Questions:**

1. The paper's two main SOTA achievements are based on two different SONAR versions: SONAR2 for video captioning and SONAR1 for multilingual v-LCM . Given that SONAR2 is reported to be substantially stronger , does this imply that the multilingual performance of v-LCM (based on SONAR1) is significantly underestimated? What is the expected performance ceiling for v-LCM if it were aligned and trained with SONAR2?

2. v-LCM is based on a latent diffusion model (LCM) , which requires iterative denoising for generation. This process is typically much slower than the autoregressive transformers used in baseline VLMs. What is the actual inference latency of v-LCM compared to models like Qwen-VL or PLM-8B, and why was this critical discussion of computational cost omitted when assessing its practical utility as a "new paradigm"?

3. v-LCM achieves success in 61 languages , yet its alignment data (12M images + 2M videos ) is likely English-dominated. How does the paper prove that v-LCM learned a true alignment between visual concepts and non-English concepts, rather than simply "inheriting" this capability as a "free lunch" from the pre-trained SONAR space?

4. Appendix B shows that contrastive loss was superior for retrieval (52.4 vs. 49.0), while the chosen MSE loss was only marginally better for captioning (38.9 vs. 38.6). Why does the more common contrastive loss, which improves retrieval alignment, appear to hurt the performance of generative tasks like captioning? Can a deeper technical analysis be provided?

5. Why did v-LCM fail to outperform baselines in Dutch and lag significantly on English tasks like VisualMRC (Table 5)? Does this expose a fundamental limitation of the v-LCM framework in handling specific languages or task types?

---

> ### Author Response · Authors · 2025-11-21
> **Response to reviewer wU7h**
>
> We would like to thank the reviewer for the thorough review. Please find below our answers to your questions.
>
> **1. Inconsistent SONAR Versions & Performance Ceiling (R1 & Q1)**
>
> > **Review:** Does this imply that the multilingual performance of v-LCM on SONAR1 (based on SONAR1) is significantly underestimated? What is the expected performance ceiling for v-LCM if it were aligned and trained with SONAR2?
>
> **Response:** We acknowledge the confusion and appreciate the opportunity to clarify. The two SONAR versions are architecturally identical (same dimension, encoder-decoder structure), but SONAR2 utilizes an improved training recipe yielding substantially better performance. As the diffusion backbone (LCM) compatible with SONAR2 is not yet available, we explored both variants:
>
> 1. **SONAR2 was used only for proving the feasbility of V-SONAR alignment pipeline for video captioning tasks (Table 2 & 3)**. However, we would like to emphasis that even with SONAR1, performance remains competitive. On PVD-Bench, SONAR1 achieves **35 R-L** (vs. second-best 31.2), and on DREAM-1K, it achieves **21 R-L** (vs. second-best 20.4). Changing versions does not qualitatively alter our conclusions.
> 2. **We then use SONAR1 to verify for leveraging it with LCM because the original LCM backbone was trained exclusively on SONAR1 embeddings**. As SONAR2 is a drop-in replacement for SONAR1 with improved performance, we do expect v-LCM performance is underestimated as you said. However, re-training of LCM with SONAR2 requires the extensive pre and post-training from scratch, which is over our compute budget for this project. But it is one of our future works.
>
> **Does using SONAR1 underestimate performance?**
> Yes. Since SONAR2 is a drop-in replacement with improved text–text similarity metrics (Table 1), we expect V-LCM’s performance to improve if retrained with SONAR2. Based on our captioning results (Appendix B and Figure 4), where SONAR2 shows an average gain of ~5 points over SONAR1, we estimate a similar performance ceiling increase for V-LCM.
>
> Finally, we will release alignment scripts to allow users to reproduce results.
>
> **2. Inference Latency (R2 & Q2)**
>
> > **Review:** What is the actual inference latency of v-LCM compared to models like Qwen-VL or PLM-8B?
>
> **Response:** LCM and V-LCM operate in a latent space where a single embedding corresponds to a sentence-level sequence in token LLMs or whole frames in VLMs. Conceptually, Large Concept Models enjoy efficiency advantages for sequence modeling [1].
>
> We measured wall-clock latency (seconds) on a single A100 GPU:
>
> | Model | VIST | WINOGROUND | Image-Para-Cap | MSRVTT-QA | MSRVTT |
> | :--- | :--- | :--- | :--- | :--- | :--- |
> | **PLM-8B** | 7.27 | 8.18 | 8.09 | 10.67 | 10.09 |
> | **vLCM** | 11.94 | 12.12 | 11.02 | 10.64 | 12.46 |
>
> **Analysis:** Without specific inference optimization, V-LCM shows a slight increase in latency due to the iterative diffusion process. However, as sequence length increases, either via video inputs (MSRVTT, MSRVTT-QA) or longer output requirements (Paragraph Captioning), the gap tends to decrease.
>
> *[1] Large Concept Models: Language Modeling in a Sentence Representation Space, arXiv 2024.*
>
> **3. Data Bias and Multilingual Generalization (R3)**
>
> > **Review:** It is unclear if this capability is learned during alignment or simply a "free lunch" inherited from the pre-trained SONAR space... How does the paper prove that v-LCM learned a true alignment... rather than simply "inheriting" this?
>
> **Response:**
> 1.  **Alignment data is not used for improving vLCM's multilingual generalization**. We only use the English data as it is the most scalable source for image/video captioning data. During alignment, Perception Encoder learns to project visual inputs into the SONAR latent manifold where textual semantics for 200 languages already reside in.
> 2.  **We attribute the multilingual generalization to vLCM's design choice**. vLCM is operating in the language/modality-agnostic space SONAR. Its design allows zero-shot transfer for vision-instruction ability to any languages can be decoded by SONAR, consequently.

---

> > ### Author Response · Authors · 2025-11-21
> >
> > **4. Performance on English Benchmarks & Specific Failures (R4)**
> >
> > > **Review:** v-LCM is not the overall best on competitive English benchmarks.
> >
> > **Response:**
> > *   The core strength of V-LCM is its generalization across 62 rich-to-low-resource languages, where it outperforms competitors on 61 languages. This breadth comes from a design that prioritizes a general, modality/language-agnostic representation space over peak performance on SoTA models. Additionally, we do not expect vLCM to surpass SoTA model in most of English, LCM itself has not gone through extensive and carefully enginneering-optimized post-training such as Qwen.
> > *   While V-LCM trails PLM-8B by 1–2 ROUGE on VisualMRC and VIDEOXUM, it achieves near-parity on video captioning and substantially higher performance on multilingual QA and captioning (+15 ROUGE in low-resource languages). We consider this as an acceptable tradeoff to acquire a better multilingual results.
> >
> > > **Review:** Why did v-LCM fail to outperform baselines in Dutch and lag significantly on English tasks like VisualMRC (Table 5)?
> > *   **The v-LCM outperforms all other models on all languages for 5 out of 6 tasks**: MSRVTT, MSRVTT-QA, OKVQA, VIST and VQA-V2, The only exception being the Thai language in VQA-v2. This may be due to low performance of SONAR1 on Thai (sentence segmentation is challenging for Thai which has no end-of-sentence punctuation). We expect a SONAR2-based LCM to mitigate this. We would like to argue that this single exception can't be seen as an evidence of a fundamental limitation.
> > On the other hand, the v-LCM has indeed mixed performance on ImageNet (discussed in Appendix F). We attribute this to ImageNet's widespread use and extensive coverage in existing VLMs training.
> > *   For VisualMRC, the task may differ from the alignment data's domain: document-style QA involves OCR text layouts, not natural visual scenes. Since V-SONAR is aligned on natural imagery, it lacks exposure to document-level features. We acknowledge this limitation and plan to extend alignment data to include document images (e.g., DocVQA, TextVQA) in future work.
> >
> > **5. MSE vs. Contrastive Loss Analysis (R5)**
> >
> > > **Review:** Why does the more common contrastive loss, which improves retrieval alignment, appear to hurt the performance of generative tasks like captioning?
> >
> > **Response:** Contrastive and MSE objectives optimize different geometrical properties of the embedding space, so a trade-off is expected:
> >
> >
> > 1. Contrastive loss optimizes relative (cosine) geometry. It explicitly pushes mismatched pairs apart while pulling matched pairs together in terms of cosine similarity. That tends to increase pairwise margin and makes nearest-neighbour retrieval easier (hence the R@1 gain). It is largely invariant to absolute vector offsets and can increase angular separation (or anisotropy) of vectors.
> >
> > 2. MSE loss optimizes absolute reconstruction in the target (SONAR) embedding space. MSE penalizes Euclidean distance to the frozen SONAR vectors, encouraging the vision embeddings to lie close in absolute position to the SONAR decoder’s training manifold. Because our downstream generation (SONAR decoder / LCM diffusion) depends on predicting/reconstructing SONAR embeddings (i.e., absolute embedding values and their distribution), minimizing Euclidean distance directly tends to preserve the statistics (norms, relative positions, local neighborhoods) that the decoder expects — which helps captioning.
> >
> > 3. Decoder sensitivity / manifold mismatch. Contrastive training can move vision vectors off the exact SONAR manifold (it only enforces relative ordering via cosine margins), producing embeddings whose norms or local covariance differ from those the SONAR decoder (and the LCM denoiser) were trained on. That small manifold shift can degrade generative reconstruction even if retrieval (based on cosine ranking) improves.
> >
> > To empirically support these arguments, we add an analysis in the Appendix B:
> >
> > |                 | vSONAR Norm | vSONAR Trace | vSONAR Volume    | Alignment Consistency (by cosine) | Alingment Consistency (by mse) |
> > |-----------------|-------------|-------|-----------|-----------------------------------|--------------------------------|
> > | MSE             |        1.22 |  0.48 | -10007.16 |                              0.41 |                           0.32 |
> > | MSE+Contrastive |        1.30 |  1.74 |  -8107.94 |                              0.31 |                           0.13 |
> >
> > It is a clear observation that **contrastive loss pushes the vSONAR embeddings to be a more expanded distribution** compared to the MSE-only (higher values in Norm, Covariance trace, and Volume). **However, the poor alignment consistency for contrastive loss**, where we calculate the correlation between vision and text similarity ranking for each sample, suggests contrastive loss breaks the local covariance structure and may break the alignment with SONAR manifold.

---

### Meta-Review · Area_Chair_gDcr · 2026-01-02

**Summary:**

This paper proposes v-SONAR that extends the multilingual SONAR to the image and video modality. The training is conducted with coarse-to-fine curriculum over three stages of vision captioning data. Another extension v-LCM, a vision-language instruction fine-tuned latent diffusion language model, further shows its improvement in multiple downstream vision-language tasks. Extensive experiments show strong performance on image/video captioning, visual question answering, and other generation tasks.

The main strengths are (1) the multilingual VLMs alignment approach is computationally efficient and scalable, (2) the proposed v-SONAR/v-LCM achieves outstanding zero-shot and instruction-following performance across 61+ languages, significantly outperforming strong baselines, (3) the evaluations on multiple tasks and ablation studies including retrieval, captioning, multilingual instruction following, and long-video tasks are comprehensive. The major weaknesses and concerns are (1) the contribution of vision-language alignment for an existing model is incremental, (2) the presevation of spatial information, object relations, and fine-grained details beyond semantic captions is not clear, which is further addressed by the rebuttal, (3) controlled baselines and justification for MSE loss are lacked, which are provided in the rebuttal.

Overall, the rebuttal addressed most of the concerns and provided comprehensive analysis and results to address reviewers' concerns, and Reviewer wLL2 increased their score from 6 to 8. The empirical contributions are significant to vision-language tasks. The remaining main weakness is the research contribution of vision-language alignment, which lacks deep analysis of the challenges and novelty compared to previous vision-language alignment methods, making the contributions more experimental. The final recommendation is accept.

**Reviewer Concerns:**

Most of the concerns are well addressed by the rebuttal with new visual commonsense reasoning experiments, Covariance Trace and LogDet analysis,extra retrieval comparison, and new latency comparison. The remaining weakness is the research contribution of vision-language alignment, which lowers the final rating.

**Reviewer Scores:**

Reviewer wLL2 agreed to increased their score from 6 to 8. The area chair thinks that the remaining negative rating would be changed from 4 to 6, which would lead to all positive ratings.

---

### Decision · Program_Chairs · 2026-01-26

Accept (Poster)